# HELMET: How to Evaluate Long-Context Language Models Effectively and Thoroughly

**Howard Yen**[P] **Tianyu Gao**[P] **Minmin Hou**[i] **Ke Ding**[i]
**Daniel Fleischer**[i] **Peter Izsak**[i] **Moshe Wasserblat**[i] **Danqi Chen**[P]
[P]Princeton Language and Intelligence, Princeton University   [i]Intel
{hyen,tianyug,danqic}@cs.princeton.edu
https://princeton-nlp.github.io/HELMET

## Abstract

Many benchmarks exist for evaluating long-context language models (LCLMs), yet developers often rely on synthetic tasks such as needle-in-a-haystack (NIAH) or an arbitrary subset of tasks. However, it remains unclear whether these benchmarks reflect the diverse downstream applications of LCLMs, and such inconsistencies further complicate model comparison. We investigate the underlying reasons behind these practices and find that existing benchmarks often provide noisy signals due to limited coverage of applications, insufficient context lengths, unreliable metrics, and incompatibility with base models. In this work, we introduce **HELMET** (**H**ow to **E**valuate **L**ong-context **M**odels **E**ffectively and **T**horoughly), a comprehensive benchmark encompassing seven diverse, application-centric categories. We also address several issues in previous benchmarks by adding controllable lengths up to 128K tokens, model-based evaluation for reliable metrics, and few-shot prompting for robustly evaluating base models. Consequently, we demonstrate that HELMET offers more reliable and consistent rankings of frontier LCLMs. Through a comprehensive study of 59 LCLMs, we find that (1) synthetic tasks like NIAH do not reliably predict downstream performance; (2) the diverse categories in HELMET exhibit distinct trends and low correlations with each other; and (3) while most LCLMs achieve perfect NIAH scores, open-source models significantly lag behind closed ones when tasks require full-context reasoning or following complex instructions—the gap widens as length increases. Finally, we recommend using our RAG tasks for fast model development, as they are easy to run and better predict other downstream performance; ultimately, we advocate for a holistic evaluation across diverse tasks.[1]

## 1 Introduction

Long-context language models (LCLMs) unlock a myriad of applications, from summarizing long documents to learning new tasks on the fly with thousands of examples. Many recent benchmarks have sought to evaluate language models' long-context abilities (Zhang et al., 2024b; An et al., 2024; Shaham et al., 2023; Bai et al., 2024, *inter alia*). However, recent developments in long-context processing (Chen et al., 2023; Xiong et al., 2023; Peng et al., 2024; Fu et al., 2024) still rely either on perplexity or on synthetic needle-in-a-haystack tasks (NIAH; Kamradt, 2024; Hsieh et al., 2024). Frontier LCLMs (Dubey et al., 2024; Team et al., 2024c; OpenAI, 2023; Team

Table 1: Most LCLMs evaluate on synthetic (Syn.) tasks. [b]: base models. ∞: ∞BENCH.

| Model | Syn. | PPL | ∞ | | ZeroSCROLLS | | | | | RAG | ICL |
|---|---|---|---|---|---|---|---|---|---|---|---|
| | | | QA | All | NQA | QS | QL | SQ | All | | |
| Gemini-1.5 | ✓ | ✓ | ✗ | ✗ | ✗ | ✗ | ✗ | ✗ | ✗ | ✗ | ✓ |
| GPT-4 | ✓ | ✗ | ✗ | ✗ | ✗ | ✗ | ✗ | ✗ | ✗ | ✗ | ✗ |
| Claude-3.5 | ✓ | ✗ | ✗ | ✗ | ✗ | ✗ | ✗ | ✗ | ✗ | ✗ | ✗ |
| Llama-3.1 | ✓ | ✗ | ✓ | ✗ | ✗ | ✓ | ✓ | ✓ | ✗ | ✗ | ✗ |
| Phi-3 | ✓ | ✗ | ✗ | ✗ | ✗ | ✓ | ✓ | ✗ | ✗ | ✗ | ✗ |
| Jamba-1.5 | ✓ | ✗ | ✓ | ✗ | ✗ | ✗ | ✗ | ✗ | ✗ | ✗ | ✗ |
| Qwen2 | ✓ | ✗ | ✗ | ✗ | ✗ | ✗ | ✗ | ✗ | ✗ | ✗ | ✗ |
| Command R | ✓ | ✗ | ✗ | ✗ | ✗ | ✗ | ✗ | ✗ | ✗ | ✓ | ✗ |
| Xiong et al. | ✗ | ✗ | ✗ | ✗ | ✓ | ✓ | ✓ | ✓ | ✓ | ✗ | ✗ |
| Chen et al.[b] | ✓ | ✓ | ✗ | ✗ | ✗ | ✗ | ✗ | ✗ | ✗ | ✗ | ✗ |
| Peng et al.[b] | ✓ | ✓ | ✗ | ✗ | ✗ | ✗ | ✗ | ✗ | ✗ | ✗ | ✗ |
| Fu et al.[b] | ✓ | ✓ | ✓ | ✗ | ✗ | ✗ | ✗ | ✗ | ✗ | ✗ | ✗ |

---

[1]Our data and code are available at https://github.com/princeton-nlp/HELMET.

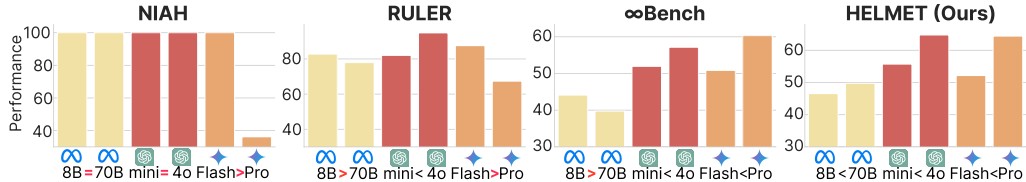

Figure 1: Long-context benchmark results of frontier LCLMs (Llama-3.1 8B/70B, GPT-4o-mini, GPT-4o-08-06, and Gemini-1.5 Flash/Pro) at 128K input length. NIAH is saturated for almost all models; RULER (Hsieh et al., 2024) and ∞BENCH (Zhang et al., 2024b) show unexpected trends for Llama-3.1 (Dubey et al., 2024). In contrast, HELMET demonstrates more consistent rankings of these frontier models.

et al., 2024a) also mostly report NIAH, sometimes with arbitrary subsets of other datasets, as shown in Table 1. Such inconsistencies complicate comparisons between different models. It is also unclear whether synthetic NIAH tasks or the few chosen datasets offer a holistic picture of models' long-context abilities and their performance in real-world applications.

*Why don't model developers agree on these evaluations*? We take a closer look and find that existing benchmarks suffer from many critical design flaws, including:

- **Insufficient coverage of downstream tasks**: Existing benchmarks either focus on synthetic tasks (Hsieh et al., 2024) or include only simple question answering (Zhang et al., 2024b). Other works study particular aspects of LCLMs, such as summarization (Chang et al., 2024), in-context learning (Li et al., 2024c), and retrieval-augmented generation (RAG; Lee et al., 2024), but they do not provide a holistic evaluation of LCLMs.
- **Inadequate lengths**: Most natural language datasets in existing benchmarks (Shaham et al., 2023; An et al., 2024; Table 4) are too short to effectively test frontier long-context abilities (usually ≥128K).
- **Unreliable metrics**: For commonly used long-document QA and summarization tasks, most existing benchmarks still rely on metrics like ROUGE (Lin, 2004), which are often noisy and unreliable (Goyal et al., 2023; Deutsch et al., 2022; Chang et al., 2024).
- **Incompatibility with base models**: Many LCLM developments focus on base models without instruction tuning, but most existing benchmarks require models to be instruction-tuned—hence developers can only rely on synthetic tasks or perplexity.

Consequently, existing benchmarks are either not applicable to long-context works (inadequate lengths or incompatibility) or provide highly noisy signals (insufficient coverage or unreliable metrics). We summarize the shortcomings of existing benchmarks in Table 2. For the three benchmarks that support a 128K[2] context length (NIAH, RULER, and ∞BENCH), we use them to evaluate six frontier models and report the numbers in Figure 1. We see that NIAH does not reflect differences across models; RULER and ∞BENCH show unexpected trends—on RULER, Gemini Flash outperforms Gemini Pro; on ∞BENCH, the 70B Llama model underperforms compared to the 8B one. This raises concerns about the reliability of the benchmarks and how they can differentiate long-context models, which likely contributes to the lack of use of these benchmarks in model development.

To address these challenges, we present **HELMET** (**H**ow to **E**valuate **L**ong-context **M**odels **E**ffectively and **T**horoughly). We curate a *diverse* set of application-centric long-context tasks across seven categories. Beyond widely adopted categories like *synthetic recall*, *long-document question answering (QA)*, and *summarization*, we also add *many-shot in-context learning* (ICL), *retrieval-augmented generation* (RAG), *passage re-ranking*, and *generation with citations* (Gao et al., 2023). We further address the shortcomings of existing benchmarks: (1) we ensure all datasets support input lengths of 128K tokens and are easily extendable to longer contexts (§2.1); (2) we introduce reference-based model evaluation for QA and summarization and show that it significantly improves over n-gram overlap metrics (§2.2); (3) we refine the prompts and in-context demonstrations used for all tasks, reducing evaluation noise caused by different output formats and allowing base models to

---

[2]Throughout the paper, we use binary prefixes K= $2^{10}$.

Table 2: Comparison of long-context benchmarks: ZeroSCROLLS (Shaham et al., 2023), Long-Bench (Bai et al., 2024), L-Eval (An et al., 2024), RULER (Hsieh et al., 2024), $\infty$BENCH (Zhang et al., 2024b), and our HELMET. $L$: input tokens. $^\dagger$: All datasets have $L < 128K$ except one dataset. $^\ddagger$: L-Eval uses LLMs to compute reference-free, pairwise win-rates; we design reference-based model evaluation for specific tasks.

| | Type of tasks | | | | | | | Benchmark features | | |
|---|---|---|---|---|---|---|---|---|---|---|
| | Cite | RAG | Re-rank | Long-QA | Summ | ICL | Synthetic Recall | Robust Eval. | $L \geq 128k$ | Controll-able $L$ |
| ZeroSCROLLS | ✗ | ✗ | ✗ | ✓ | ✓ | ✗ | ✗ | ✗ | ✗$^\dagger$ | ✗ |
| LongBench | ✗ | ✓ | ✗ | ✓ | ✓ | ✓ | ✓ | ✗ | ✗$^\dagger$ | ✗ |
| L-Eval | ✗ | ✓ | ✗ | ✓ | ✓ | ✗ | ✗ | ✓$^\ddagger$ | ✗$^\dagger$ | ✗ |
| RULER | ✗ | ✗ | ✗ | ✗ | ✗ | ✗ | ✓ | ✓ | ✓ | ✓ |
| $\infty$BENCH | ✗ | ✗ | ✗ | ✓ | ✓ | ✗ | ✓ | ✗ | ✓ | ✓ |
| HELMET (Ours) | ✓ | ✓ | ✓ | ✓ | ✓ | ✓ | ✓ | ✓ | ✓ | ✓ |

be evaluated robustly across most categories (§2.3). Together, HELMET enables a holistic evaluation of long-context capabilities and provides more reliable signals for model development. Figure 1 demonstrates that HELMET can clearly differentiate models with varying capabilities and reflect comparisons consistent with human perception. More discussion on HELMET's improvements over previous benchmarks and direct comparisons can be found in §A.

To understand the progress of LCLMs and how different long-context capabilities correlate with one another, we evaluate a comprehensive list of 59 LCLMs of various architectures, scales, and training approaches. Our analysis reveals that (1) synthetic tasks poorly indicate ownstream performance (§3.1), (2) different categories in HELMET show distinct trends (§3.2), and (3) open-source models significantly lag behind closed ones on tasks that require reasoning over long contexts or following complex instructions—the gap further widens as context length increases (§3.3). Finally, we find that the RAG category strikes a good balance between ease of use, stronger correlation with downstream tasks, and compatibility with base models. Ultimately, it is imperative to evaluate LCLMs across a diverse spectrum of categories. We hope that our insights provide a more effective way to evaluate LCLMs for future model development and benchmarking.

## 2  OUR BENCHMARK: HELMET

In this work, we seek to overcome the shortcomings of existing benchmarks by meeting the following desiderata: (1) diverse coverage across different tasks and capabilities of LCLMs, (2) controllable context lengths that support more than 128K input tokens, and (3) reliable evaluation for both base and instruction-tuned models. In this section, we describe the datasets used in HELMET and how they improve upon existing evaluation benchmarks in terms of settings and metrics. An overview of HELMET is shown in Table 3.

### 2.1  REALISTIC AND DIVERSE LONG-CONTEXT APPLICATIONS

**Retrieval-augmented generation (RAG).** We use open-domain question answering (ODQA)—which requires retrieving from a knowledge corpus and then generating correct answers (Chen et al., 2017)—as a representation of retrieval-augmented generation (RAG) applications. We utilize Natural Questions (NQ; Kwiatkowski et al., 2019), TriviaQA (TQA; Joshi et al., 2017), HotpotQA (Yang et al., 2018), and PopQA (Mallen et al., 2023). We use the gold passage (the passage with the answer) from Petroni et al. (2021), or otherwise select any passage that contains the answer.

Given an input length $L$, we first determine the number of passages $k$ that can fit within $L$ tokens, then retrieve $k$ passages[3] from the corpus[4] that *do not contain the answer* as *distractors*. This differs from previous works that *randomly sample passages* from the corpus (Lee et al., 2024) and is more

---

[3]We use `Alibaba-NLP/gte-large-en-v1.5` for retrieval (Zhang et al., 2024a).
[4]We use Wikipedia 2019-8-01 dump, split into 100-word passages (Petroni et al., 2021).

Table 3: Overview of evaluation datasets. We select datasets that cover various important long-context capabilities. SubEM: substring exact match.

| Category | Dataset | Metrics | Description |
|---|---|---|---|
| **Retrieval-augmented generation** | Natural Questions | SubEM | Factoid question answering |
| | TriviaQA | SubEM | Trivia question answering |
| | PopQA | SubEM | Long-tail entity question answering |
| | HotpotQA | SubEM | Multi-hop question answering |
| **Generation with citations** | ALCE ASQA | Recall, Cite | Answer ambiguous questions with citations |
| | ALCE Qampari | Recall, Cite | Answer factoid questions with citations |
| **Passage re-ranking** | MS MARCO | NDCG@10 | Rerank passage for a query |
| **Many-shot in-context learning** | TREC Coarse | Accuracy | Question type classification, 6 labels |
| | TREC Fine | Accuracy | Question type classification, 50 labels |
| | NLU | Accuracy | Task intent classification, 68 labels |
| | BANKING77 | Accuracy | Banking intent classification, 77 labels |
| | CLINC150 | Accuracy | Intent classification, 151 labels |
| **Long-document QA** | NarrativeQA | Model-based | Book and movie script QA |
| | ∞BENCH QA | ROUGE F1 | Novel QA with entity replacement |
| | ∞BENCH MC | Accuracy | Novel multiple-choice QA with entity replacement |
| **Summarization** | ∞BENCH Sum | Model-based | Novel summarization with entity replacement |
| | Multi-LexSum | Model-based | Summarizing multiple legal documents |
| **Synthetic recall** | JSON KV | SubEM | Retrieve a key in JSON dictionary |
| | RULER MK Needle | SubEM | Retrieve the needle (a number) within noisy needles |
| | RULER MK UUID | SubEM | Retrieve the needle (a UUID) within noisy needles |
| | RULER MV | SubEM | Retrieve multiple values for one needle (key) |

realistic and challenging. For NQ, TQA, and PopQA, we take the top $k-1$ distractors and insert the gold passage at six evenly distributed positions following Liu et al. (2023). For HotpotQA, which requires two gold passages, we combine them and the top $k-2$ distractors and randomly shuffle them into three permutations. We use substring exact match (SubEM; whether the answer is included in the output), following previous work (Asai et al., 2024a). See §B.1 for more details.

**Generation with citations (Cite).** We leverage ALCE (Gao et al., 2023) to evaluate LCLMs on a realistic application of answering questions while providing correct attributions (Bohnet et al., 2022). Given multi-faceted questions and relevant passages, models are required to generate a long-text answer and cite supporting passage IDs at the end of each sentence. This tests models' ability to utilize the passages in the context and also to follow the instructions about citation formats. We use the ASQA (Stelmakh et al., 2022) and QAMPARI (Rubin et al., 2022) subsets from ALCE. For an input length $L$, we first determine the number of passages $k$, and use the top $k$ retrieved passages from Wikipedia as contexts. The model's outputs are evaluated on correctness and citation quality, and we report the average across all metrics. See §B.2 for more details.

**Passage re-ranking (Re-rank).** Re-ranking retrieved passages based on their relevance to the query is an important application of LCLMs (Sun et al., 2023). The task requires the model to retrieve relevant information, compare, and reason over different parts of the contexts. We use the MS MARCO dataset (Bajaj et al., 2018), where each instance contains a query and passages retrieved by BM25 (Robertson & Zaragoza, 2009) from the Internet. Each passage has annotations of a relevance label—perfect, highly relevant, or not relevant. We determine the number of passages $k$ from the input length $L$, and randomly sample $k$ passages with balanced labels for each test query. The model is prompted with the query and the shuffled $k$ passages and is instructed to output the top-10 document IDs ranked by relevance. We report NDCG@10. Details are in §B.3.

**Many-shot in-context learning (ICL).** In-context learning (ICL) is a key ability that enables LLMs to adapt to new tasks on the fly (Brown et al., 2020). Recent studies (Ratner et al., 2023; Xu et al., 2024; Li et al., 2024c; Bertsch et al., 2024) explore performing many-shot ICL (with thousands of examples) with LCLMs. Following Bertsch et al. (2024), we focus on datasets with large label spaces: TREC-coarse, TREC-fine (Li & Roth, 2002), BANKING77 (Casanueva et al., 2020), CLINC150 (Larson et al., 2019), and NLU (Liu et al., 2019). We adjust the number of shots to control the input length $L$, and the number of examples in each class is balanced. We report accuracy on the test set.

One *difference* from previous works is that we map original natural language labels (e.g., *location*) into numbered labels (i.e., *0, 1*) to test how well a model can learn new tasks instead of relying on its pre-trained priors (Wei et al., 2023; Pan et al., 2023; Min et al., 2022). More details are in §B.4.

**Long-document question answering (LongQA).** We use NarrativeQA (Kočiský et al., 2018) and the English book QA and multiple choice (MC) subsets from ∞BENCH (Zhang et al., 2024b) for evaluating long-document QA. We select those tasks for their abundant context lengths (Table 4). We truncate the document from the end based on $L$. We use ROUGE F1 for ∞BENCH QA (answers are mostly entity names) and accuracy for ∞BENCH MC. For NarrativeQA, where the answers can be long text and open-ended, we design and use a model-based evaluation (§2.2).

Table 4: Dataset lengths.

| Datasets | Medium | Max |
|---|---|---|
| ZeroSCROLLS | | |
| QASPER | 6K | 12K |
| GovReport | 12K | 33K |
| QuALITY | 9K | 11K |
| SQuALITY | 8K | 10K |
| HELMET | | |
| NarrativeQA | 73K | 518K |
| ∞BENCH QA | 191K | 835K |
| ∞BENCH MC | 167K | 835K |
| ∞BENCH Sum | 154K | 835K |
| Multi-LexSum | 90K | 5M |

**Summarization (Summ).** Summarization tests LCLMs' ability to synthesize information across the contexts. We choose Multi-LexSum (legal document summarization) and the English summarization task from ∞BENCH (novel summarization) for their extensive lengths (Table 4). We truncate the document from the end based on the evaluation length $L$. We use our model-based evaluation (§2.2) for both datasets instead of the commonly used ROUGE, as it better reflects human judgment.

**Synthetic recall.** Synthetic recall tasks, such as needle-in-a-haystack (NIAH), stress test models' ability to recall relevant information (the "needle") from long contexts. They have gained popularity for being easy to use (as they can test any arbitrary length) and easy to control (can placing the "needle" at any position). For this category, we select multiple synthetic recall tasks from RULER (an extended version of NIAH; Hsieh et al., 2024) and also add a JSON KV retrieval task (Liu et al., 2023), which we find more challenging. We intentionally select the synthetic tasks that correlate well with application-driven tasks; in-depth discussions are in §3.1. Following previous works, we report the percentage of the ground truth answers that are substrings in the generation (SubEM). Refer to §B.5 for more details.

## 2.2 RELIABLE EVALUATION METRICS

Existing long-context benchmarks (Zhang et al., 2024b; Shaham et al., 2023) largely rely on n-gram overlap metrics like ROUGE (Lin, 2004), which have been shown to correlate poorly with human judgment for tasks with long outputs, such as summarization (Goyal et al., 2023; Deutsch et al., 2022; Krishna et al., 2023). L-Eval (An et al., 2024) uses LLMs to score reference-free "win rates," which neglect the available answer annotations and always require evaluating model pairs. Instead, we design a *reference-based model evaluation* method for long-document QA and summarization that is more reliable and easy to use.

**Question answering.** In NarrativeQA, we prompt GPT-4o[5] with the question, the ground truth, and the model output to check for fluency and correctness. The fluency score is either 0 (incoherent or repetitive) or 1 (fluent), and the correctness score takes on the value of 0 (incorrect), 1 (partly correct), 2 (correct but not fully relevant), and 3 (correct and relevant). We take the product of the two as the final score, normalizing it to a range of $[0, 100]$.

**Summarization.** Following previous works (Kamoi et al., 2023; Zhang & Bansal, 2021), we first decompose the gold summary into atomic claims and use GPT-4o to check if each claim is supported by the generation (recall) and if each sentence in the generation is supported by the reference summary (precision). We then compute the F1 score from the recall and precision scores. Additionally, we ask GPT-4o to evaluate fluency (0 or 1) and take its product with the F1 score as the final score. In each step, we prompt GPT-4o with handwritten examples.

---

[5] GPT-4o-2024-05-13

| | Multi-LexSum | | | | | | | | | | InfBench Sum | | | | | | | | | |
|---|---|---|---|---|---|---|---|---|---|---|---|---|---|---|---|---|---|---|---|---|
| | ROUGE ? | | | | | Ours 💡 | | | | | ROUGE ? | | | | | Ours 💡 | | | | |
| GPT-4o-08 | 22.2 | 22.7 | 23.5 | 23.6 | 23.8 | 43.0 | 47.9 | 51.8 | 51.3 | 53.5 | 15.9 | 16.7 | 17.0 | 17.3 | 17.0 | 13.6 | 20.7 | 26.0 | 28.9 | 32.9 |
| Gemini-1.5-Pro | 23.5 | 24.3 | 24.7 | 24.9 | 25.1 | 42.8 | 45.0 | 47.9 | 52.4 | 58.3 | 14.9 | 15.6 | 16.2 | 17.2 | 17.0 | 17.2 | 20.5 | 24.8 | 31.2 | 34.5 |
| Claude-3.5-Sonnet | 22.5 | 20.7 | 20.4 | 20.2 | 19.8 | 43.8 | 41.8 | 41.8 | 42.1 | 44.0 | 14.2 | 16.0 | 16.9 | 14.6 | 14.5 | 14.1 | 25.9 | 31.2 | 29.1 | 29.2 |
| Llama-3.2-1B-Inst | 21.1 | 22.0 | 22.2 | 22.2 | 22.0 | 15.2 | 16.8 | 20.8 | 18.5 | 15.1 | 14.1 | 15.5 | 15.8 | 16.5 | 16.4 | 0.9 | 1.9 | 2.4 | 2.7 | 3.1 |
| Llama-3.2-3B-Inst | 20.9 | 20.3 | 21.6 | 22.8 | 23.3 | 33.1 | 34.2 | 35.1 | 39.1 | 39.7 | 15.5 | 16.1 | 16.5 | 16.3 | 16.4 | 4.1 | 8.2 | 11.9 | 11.4 | 12.1 |
| Llama-3.1-8B-Inst | 24.5 | 24.6 | 23.5 | 23.8 | 24.7 | 38.8 | 39.3 | 42.7 | 39.4 | 38.8 | 15.5 | 16.2 | 16.4 | 16.9 | 17.3 | 3.7 | 9.2 | 12.3 | 13.9 | 15.3 |
| Llama-3.1-70B-Inst | 23.3 | 23.4 | 23.7 | 23.8 | 23.6 | 41.4 | 45.9 | 47.5 | 44.3 | 43.9 | 15.9 | 16.5 | 16.9 | 17.0 | 17.7 | 10.8 | 16.8 | 20.8 | 21.5 | 19.3 |
| Mistral-7B-Inst-v0.1 | 23.1 | 22.1 | 22.3 | 21.4 | 21.2 | 25.5 | 21.2 | 17.8 | 19.1 | 15.9 | 11.5 | 12.3 | 12.6 | 12.6 | 12.4 | 1.5 | 3.1 | 2.5 | 2.9 | 2.5 |
| Mistral-7B-Inst-v0.3 | 21.9 | 22.6 | 23.2 | 20.0 | 15.6 | 33.3 | 33.3 | 32.3 | 19.8 | 10.0 | 14.4 | 15.0 | 15.0 | 13.7 | 11.9 | 9.2 | 10.2 | 11.0 | 1.8 | 0.2 |
| | 8K | 16K | 32K | 64K | 128K | 8K | 16K | 32K | 64K | 128K | 8K | 16K | 32K | 64K | 128K | 8K | 16K | 32K | 64K | 128K |

Figure 2: Comparison between ROUGE-L F1 and our model-based evaluation metric on summarization tasks. Our metric shows more consistent trends: it reflects the performance gain on GPT-4o with increased input length, while ROUGE remains almost the same; our metric also clearly differentiates models while ROUGE shows little distinction.

Empirically, our reference-based model evaluation reflects more consistent trends, as shown in Figure 2: (1) Llama-3.1-8B-Inst achieves similar ROUGE scores to GPT-4o, while our evaluation reveals a significant gap. (2) Our metric better identifies incoherent generations and shows lower performance for models with smaller context windows, such as the Mistral models. (3) Our metric exhibits a substantially more positive trend for GPT-4o as input length increases, whereas ROUGE-L remains within a 2-point absolute difference. We further validate the model-based evaluation through human studies, which suggest our new metrics strongly correlate with human judgments: for example, on ∞BENCH Sum, our metric reaches a human-model agreement of Cohen's $\kappa = 0.91$ for summary precision and $\kappa = 0.76$ for recall. More details on the human studies are in §B.

## 2.3 ROBUST PROMPTING AND CONTROLLED EVALUATION SETTINGS

**Robust prompting reduces noise and enables evaluation on base models.** Many long-context benchmarks require models to follow instructions and only support evaluating instruction-tuned models (Zhang et al., 2024b; Shaham et al., 2023). However, many model developments do not incorporate instruction tuning (Chen et al., 2023; Fu et al., 2024), leaving these models reliant on perplexity-based evaluation or synthetic tasks. To support long-context research efforts, we design our benchmark so that at least a subset of the datasets accommodates evaluating base models.

Existing benchmarks mostly use zero-shot prompting (Shaham et al., 2023; Zhang et al., 2024b), which leads to inconsistent output formats, especially for base models. For example, the model may output a long answer in RAG when a short answer is required. We add two-shot demonstrations in the prompt for all tasks to address this problem.[6] For long-document QA and summarization, we replace the original document with a placeholder phrase to reduce the number of input tokens in the ICL example. As shown in Table 8, both base and instruction-tuned models significantly benefit from the demonstrations.

Furthermore, we employ the length-instruction-enhanced evaluation from L-Eval for long-generation tasks (i.e., summarization), which has been shown to have substantially more consistent and reliable evaluations (An et al., 2024). As a result, we find that our reproduction of previous datasets, such as ∞BENCH QA tasks, better reflects the capabilities of LCLMs, as shown in Table 7. The use of demonstrations and improved instructions more accurately depicts how models perform in real applications.

**Controlled input length and difficulty.** An important dimension to consider when evaluating LCLMs is the input length $L$, as longer inputs can provide more information while challenging the model's ability to process distracting contexts. As we discussed in §2.1, we can control the input length $L$ for each task by either adjusting the number of retrieved passages, the number of demonstrations, or truncating the document text to fit within the specified lengths. This allows us to study model performance at or beyond the length of current frontier LCLMs ($\geq$ 128K).

---

[6]Except for ICL (the number of shots varies) and RULER (we follow the original formatting).

As shown in Figure 1 and 2, HELMET provides rankings more consistent with human perception of model performance. The diverse tasks, controllable lengths, and robust evaluation metrics and prompting enable a thorough examination of LCLMs across multiple dimensions.

## 3 ANALYSIS

We evaluate 59 LCLMs with HELMET. To our best knowledge, this is the most thorough and controlled comparison of long-context models on diverse applications. These models cover closed-source models, such as GPT-4, Claude, and Gemini, as well as open-source model families, such as Llama (Dubey et al., 2024), Mistral (Jiang et al., 2023), Phi (Abdin et al., 2024), and Qwen (Qwen et al., 2025). We also consider models that use different architectures—full-attention transformers (Vaswani et al., 2017), sliding-window attention (Beltagy et al., 2020), and hybrid models with SSM modules (Dao & Gu, 2024). We also benchmark position extrapolation models such as YaRN (Peng et al., 2024) and LongRoPE (Ding et al., 2024). We list all the models evaluated in Table 15. We evaluate each model at input lengths: $L \in \{8K, 16K, 32K, 64K, 128K\}$, where $L$ is the number of Llama-2 tokens (Touvron et al., 2023), and use greedy decoding for all models to ensure consistency. We randomly sample 100 to 600 examples from each dataset; more details are in §D.

### 3.1 SIMPLE SYNTHETIC TASKS ARE POOR PREDICTORS OF REAL-WORLD PERFORMANCE

Many model developers rely on simple synthetic tasks, such as NIAH, for evaluating long-context language models, but it is unclear if these tasks accurately represent real-world performance. To this end, we calculate Spearman's rank correlation $\rho$ between synthetic and real-world tasks for 35 instruction-tuned models. First, Figure 3 shows that none of the synthetic tasks achieves an average correlation higher than 0.8. We also make the following observations.

**Not all synthetic tasks are created equal.** The original NIAH, which places a needle in the middle of unrelated essays and asks the model to retrieve it, exhibits weak correlation with real-world tasks: all correlations are $\leq 0.8$. Similarly, the popular RULER average score—which includes not only NIAH variants but also synthetic aggregation, multi-hop tracing, and QA—does not yield strong correlations (all $< 0.85$).

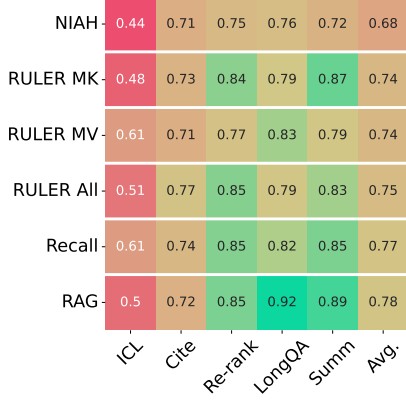

Figure 3: Spearman's rank correlation at 128K input length, calculated across 35 instruction-tuned models.

We take a closer look at different RULER tasks and find that harder recall-type tasks are more reflective of real-world categories—for example, RULER MK, which places distracting needles around the target needle. Despite the overall low correlation, we believe these tasks can still serve as a useful sanity check during model development. We compile several such RULER tasks, along with JSON KV, to form the HELMET synthetic recall set (more discussions in §E.1).

**Tasks with noisier, more distracting contexts better differentiate models.** To understand why synthetic tasks exhibit weak correlation with real-world tasks, we plot the performance of different models on NIAH, RULER MK (one of our recall tasks), and HotpotQA (one of our RAG tasks) in Figure 4. We use ∞BENCH QA as a representative real-world task. We find that most models achieve either perfect or near-zero performance on the original NIAH, leaving few data points in the middle and resulting in poor separability between models. In contrast, RULER MK, which introduces more distracting contexts, better distributes model performance between 0% and 100%, leading to clearer differentiation.

**RAG is a better proxy for real-world tasks.** Finally, we find that RAG datasets, such as HotpotQA, consistently achieve higher correlation with other real-world tasks. Figure 4 also shows that HotpotQA exhibits an almost linear relationship with the QA dataset. Similar to synthetic tasks, RAG

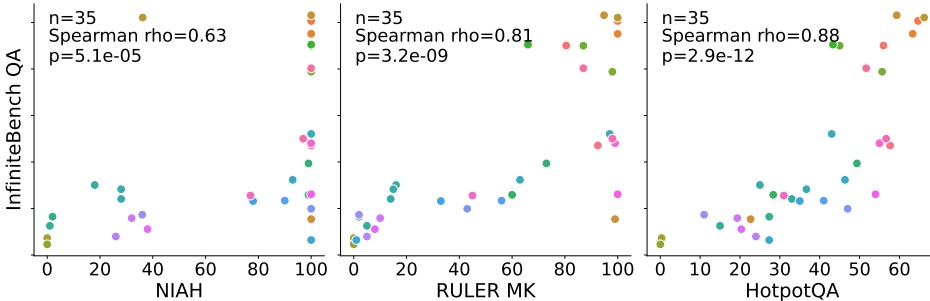

Figure 4: Distribution of instruction-tuned models' performance on ∞BENCH QA with respect to NIAH, RULER MK, and HotpotQA.

tasks are easy to control and assess models' recall abilities. However, since all passages are retrieved and relevant to the query, RAG contexts are more distracting and therefore harder to saturate.

## 3.2 DIVERSE LCLM APPLICATIONS CALL FOR DIVERSE EVALUATION

In long-context language modeling, realistic tasks are often only used in isolation (Karpinska et al., 2024; Li et al., 2024c; Dubey et al., 2024), which limits the understanding of LCLMs in a broader context. In this work, we cross-examine model performance over a wide range of real tasks, and find that different categories do not consistently correlate with each other, as shown in Figure 5.

show moderate correlation due to their shared retrieval component. However, the added complexity of generating citations in ALCE results in lower correlation with other categories. Naturally, RAG and passage re-ranking moderately correlate due to the shared retrieval component As shown in Figure 9, generating correct answers and producing valid citations are not strongly correlated, suggesting that instruction following and recalling facts within long contexts are distinct capabilities.

Furthermore, some categories—generation with citations and in-context learning—do not correlate well with other categories. Intuitively, summarization tests for the model's ability to aggregate information across the entire input, while ICL evaluates its ability to learn new tasks from many examples. Such capabilities are orthogonal to recall facts in long contexts. Such capabilities are orthogonal to recalling facts in long contexts. Therefore, model developers should evaluate across these distinct axes to form a more holistic understanding of a model's capabilities (see additional analysis in §E.2).

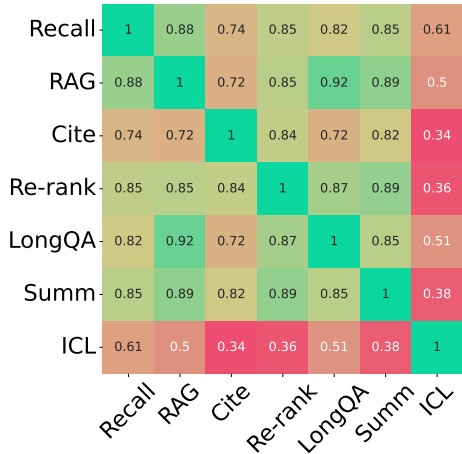

Figure 5: Spearman rank correlation between different categories at $L =$128K.

## 3.3 MODEL PERFORMANCE ACROSS TASKS AND LENGTHS

We show the performance of instruction-tuned models on HELMET at five different lengths in Figure 6, and the full results are illustrated in Figure 10. We analyze the model performance across two critical dimensions of long-context language modeling: *task complexity* and *input length*.

**Open-source models lag behind closed-source models on complex tasks.** First, we consider the performance of frontier LCLMs at the longest input length of 128K tokens. We find that the closed-source models—notably GPT-4o-08 and Gemini-1.5-Pro—stand out as the strongest LCLMs. Other than ICL, the closed-source models outperform the open-source models on all tasks. The gap is relatively small on synthetic recall and LongQA, where the task is to retrieve information from the context. There is a stark contrast in the generation with citations and re-ranking performance, where the closed-source models are 30 to 40 absolute points better than the best open-source models.

**Recall**

| | 8K | 16K | 32K | 64K | 128K |
|---|---|---|---|---|---|
| GPT-4 | 99.5 | 99.8 | 97.6 | 91.2 | 73.5 |
| GPT-4o-05 | 93.2 | 92.2 | 91.2 | 90.1 | 82.4 |
| GPT-4o-08 | 100.0 | 100.0 | 100.0 | 100.0 | 99.9 |
| GPT-4o-mini | 99.8 | 99.8 | 99.1 | 96.1 | 89.6 |
| Claude-3.5-Sonnet | 99.5 | 96.6 | 96.2 | 97.2 | 94.7 |
| Gemini-1.5-Flash | 93.5 | 93.8 | 93.7 | 93.2 | 91.2 |
| Gemini-1.5-Pro | 87.5 | 87.3 | 96.3 | 87.2 | 91.0 |
| Llama-3.1-8B | 100.0 | 100.0 | 99.7 | 99.2 | 95.2 |
| Llama-3.1-70B | 100.0 | 100.0 | 99.9 | 98.5 | 90.7 |
| Llama-3.3-70B | 100.0 | 100.0 | 99.6 | 98.5 | 81.8 |
| Mistral-Nemo | 99.9 | 95.6 | 65.6 | 30.1 | 14.6 |
| MegaBeam-Mistral | 97.8 | 97.1 | 95.9 | 94.2 | 89.6 |
| Ministral-8B | 98.9 | 97.7 | 95.1 | 30.9 | 12.8 |
| Phi-3-mini-128k | 97.8 | 96.8 | 95.1 | 95.4 | 50.1 |
| Phi-3-small-128k | 94.0 | 93.6 | 83.4 | 74.6 | 22.3 |
| Phi-3-med-128k | 91.6 | 88.9 | 79.3 | 69.3 | 24.5 |
| Phi-3.5-mini | 98.3 | 97.8 | 94.7 | 91.9 | 48.8 |
| Qwen2.5-72B | 100.0 | 100.0 | 98.4 | 71.4 | 38.4 |
| Qwen2.5-7B-1M | 96.9 | 98.5 | 94.1 | 92.8 | 86.0 |
| Qwen2.5-14B-1M | 100.0 | 99.9 | 99.4 | 99.2 | 97.8 |
| Jamba-1.5-Mini | 98.4 | 95.4 | 97.3 | 93.2 | 90.0 |
| ProLong | 99.4 | 99.7 | 99.2 | 99.4 | 98.8 |

**RAG**

| | 8K | 16K | 32K | 64K | 128K |
|---|---|---|---|---|---|
| GPT-4 | 75.1 | 73.5 | 70.9 | 68.1 | 64.7 |
| GPT-4o-05 | 72.5 | 72.4 | 72.6 | 72.7 | 70.4 |
| GPT-4o-08 | 73.9 | 74.0 | 72.4 | 71.5 | 70.2 |
| GPT-4o-mini | 72.6 | 72.1 | 70.0 | 68.7 | 68.0 |
| Claude-3.5-Sonnet | 55.4 | 35.8 | 33.4 | 32.5 | 38.1 |
| Gemini-1.5-Flash | 71.9 | 70.5 | 69.2 | 68.0 | 66.5 |
| Gemini-1.5-Pro | 73.6 | 73.4 | 72.3 | 71.8 | 71.1 |
| Llama-3.1-8B | 70.8 | 68.8 | 66.5 | 65.8 | 59.5 |
| Llama-3.1-70B | 74.0 | 71.5 | 70.9 | 69.5 | 56.2 |
| Llama-3.3-70B | 74.4 | 72.7 | 70.3 | 69.2 | 55.1 |
| Mistral-Nemo | 67.8 | 63.5 | 57.3 | 47.1 | 40.0 |
| MegaBeam-Mistral | 64.7 | 63.0 | 61.8 | 59.7 | 57.0 |
| Ministral-8B | 66.7 | 66.0 | 63.5 | 47.1 | 35.2 |
| Phi-3-mini-128k | 62.5 | 61.4 | 59.4 | 56.3 | 46.7 |
| Phi-3-small-128k | 67.1 | 64.7 | 62.6 | 58.5 | 33.8 |
| Phi-3-med-128k | 66.1 | 63.8 | 62.3 | 57.8 | 44.8 |
| Phi-3.5-mini | 61.9 | 60.6 | 57.1 | 52.5 | 43.1 |
| Qwen2.5-72B | 72.5 | 72.0 | 70.2 | 63.1 | 43.0 |
| Qwen2.5-7B-1M | 67.7 | 68.3 | 64.9 | 64.0 | 59.0 |
| Qwen2.5-14B-1M | 72.2 | 70.8 | 68.9 | 66.0 | 62.0 |
| Jamba-1.5-Mini | 65.2 | 64.1 | 63.4 | 61.8 | 57.3 |
| ProLong | 67.4 | 67.5 | 66.4 | 65.7 | 63.2 |

**Cite**

| | 8K | 16K | 32K | 64K | 128K |
|---|---|---|---|---|---|
| GPT-4 | 44.0 | 42.7 | 26.1 | 3.1 | 1.2 |
| GPT-4o-05 | 44.2 | 43.7 | 42.3 | 42.4 | 40.4 |
| GPT-4o-08 | 45.4 | 46.7 | 45.4 | 45.4 | 44.3 |
| GPT-4o-mini | 36.1 | 33.7 | 30.0 | 27.1 | 24.3 |
| Claude-3.5-Sonnet | 34.8 | 32.2 | 28.3 | 26.3 | 18.7 |
| Gemini-1.5-Flash | 48.7 | 49.4 | 48.1 | 40.2 | 31.7 |
| Gemini-1.5-Pro | 47.2 | 45.9 | 44.8 | 46.2 | 43.6 |
| Llama-3.1-8B | 36.3 | 26.7 | 10.9 | 13.1 | 2.9 |
| Llama-3.1-70B | 44.3 | 41.3 | 40.1 | 29.8 | 7.5 |
| Llama-3.3-70B | 40.4 | 38.6 | 35.7 | 31.0 | 10.9 |
| Mistral-Nemo | 32.7 | 8.2 | 2.8 | 1.1 | 0.5 |
| MegaBeam-Mistral | 22.0 | 14.1 | 9.0 | 4.5 | 4.0 |
| Ministral-8B | 36.7 | 24.1 | 7.6 | 0.7 | 0.4 |
| Phi-3-mini-128k | 23.3 | 18.5 | 10.1 | 2.1 | 0.6 |
| Phi-3-small-128k | 17.9 | 13.5 | 6.9 | 3.5 | 3.0 |
| Phi-3-med-128k | 39.8 | 26.4 | 9.7 | 5.3 | 3.3 |
| Phi-3.5-mini | 22.2 | 15.8 | 7.2 | 2.0 | 1.6 |
| Qwen2.5-72B | 42.7 | 40.8 | 39.0 | 25.1 | 8.0 |
| Qwen2.5-7B-1M | 36.2 | 17.9 | 8.6 | 9.6 | 4.8 |
| Qwen2.5-14B-1M | 41.9 | 35.4 | 32.9 | 28.2 | 27.1 |
| Jamba-1.5-Mini | 14.4 | 8.3 | 6.6 | 6.4 | 3.1 |
| ProLong | 34.6 | 27.1 | 12.8 | 5.8 | 1.4 |

**Re-rank**

| | 8K | 16K | 32K | 64K | 128K |
|---|---|---|---|---|---|
| GPT-4 | 87.8 | 78.1 | 54.7 | 25.8 | 9.6 |
| GPT-4o-05 | 86.3 | 77.9 | 67.7 | 57.7 | 47.7 |
| GPT-4o-08 | 86.9 | 80.5 | 68.5 | 58.4 | 50.0 |
| GPT-4o-mini | 80.8 | 71.1 | 58.8 | 42.0 | 31.2 |
| Claude-3.5-Sonnet | 75.2 | 41.0 | 34.8 | 17.9 | 7.2 |
| Gemini-1.5-Flash | 87.8 | 80.8 | 74.0 | 61.2 | 50.8 |
| Gemini-1.5-Pro | 87.6 | 84.1 | 76.5 | 68.4 | 59.7 |
| Llama-3.1-8B | 67.0 | 51.7 | 43.4 | 32.8 | 14.0 |
| Llama-3.1-70B | 83.5 | 73.5 | 60.2 | 41.0 | 24.5 |
| Llama-3.3-70B | 84.8 | 76.7 | 65.0 | 42.3 | 26.2 |
| Mistral-Nemo | 66.3 | 48.3 | 14.8 | 0.0 | 0.0 |
| MegaBeam-Mistral | 56.4 | 40.3 | 35.0 | 25.6 | 14.7 |
| Ministral-8B | 67.4 | 49.4 | 32.6 | 0.0 | 0.0 |
| Phi-3-mini-128k | 53.1 | 33.1 | 23.5 | 14.0 | 5.8 |
| Phi-3-small-128k | 46.1 | 35.5 | 27.4 | 16.9 | 1.9 |
| Phi-3-med-128k | 52.6 | 36.7 | 26.1 | 12.9 | 6.6 |
| Phi-3.5-mini | 51.3 | 36.4 | 25.6 | 14.2 | 7.8 |
| Qwen2.5-72B | 86.0 | 75.3 | 60.3 | 41.0 | 24.5 |
| Qwen2.5-7B-1M | 71.3 | 57.4 | 45.9 | 32.1 | 24.8 |
| Qwen2.5-14B-1M | 74.3 | 57.9 | 49.9 | 38.0 | 27.9 |
| Jamba-1.5-Mini | 62.1 | 46.3 | 33.4 | 25.4 | 14.6 |
| ProLong | 60.4 | 47.4 | 40.2 | 31.1 | 22.5 |

**LongQA**

| | 8K | 16K | 32K | 64K | 128K |
|---|---|---|---|---|---|
| GPT-4 | 46.6 | 46.1 | 46.7 | 44.2 | 46.7 |
| GPT-4o-05 | 43.8 | 48.0 | 53.6 | 54.3 | 62.0 |
| GPT-4o-08 | 41.8 | 48.1 | 50.8 | 55.5 | 59.3 |
| GPT-4o-mini | 35.3 | 39.8 | 45.8 | 49.3 | 54.8 |
| Claude-3.5-Sonnet | 29.2 | 23.1 | 27.2 | 12.6 | 12.6 |
| Gemini-1.5-Flash | 33.9 | 42.4 | 46.3 | 52.3 | 57.3 |
| Gemini-1.5-Pro | 35.5 | 41.5 | 49.3 | 52.6 | 59.6 |
| Llama-3.1-8B | 25.2 | 32.6 | 40.5 | 44.6 | 43.2 |
| Llama-3.1-70B | 31.2 | 40.0 | 47.0 | 54.0 | 56.3 |
| Llama-3.3-70B | 34.7 | 41.8 | 49.8 | 52.2 | 52.0 |
| Mistral-Nemo | 31.0 | 33.4 | 27.3 | 21.4 | 22.5 |
| MegaBeam-Mistral | 20.7 | 30.7 | 32.2 | 34.9 | 37.3 |
| Ministral-8B | 28.8 | 33.4 | 34.9 | 23.7 | 25.0 |
| Phi-3-mini-128k | 28.2 | 31.4 | 32.9 | 33.9 | 29.9 |
| Phi-3-small-128k | 25.0 | 28.5 | 36.0 | 41.8 | 27.5 |
| Phi-3-med-128k | 22.7 | 21.2 | 19.1 | 18.4 | 26.9 |
| Phi-3.5-mini | 24.7 | 26.4 | 29.3 | 30.6 | 28.5 |
| Qwen2.5-72B | 36.2 | 40.5 | 44.4 | 41.5 | 38.9 |
| Qwen2.5-7B-1M | 28.8 | 36.7 | 37.9 | 36.5 | 38.6 |
| Qwen2.5-14B-1M | 31.0 | 38.1 | 39.7 | 44.1 | 47.8 |
| Jamba-1.5-Mini | 35.9 | 40.0 | 48.1 | 51.0 | 54.2 |
| ProLong | 28.3 | 36.9 | 37.6 | 40.8 | 43.9 |

**Summ**

| | 8K | 16K | 32K | 64K | 128K |
|---|---|---|---|---|---|
| GPT-4 | 28.2 | 29.1 | 28.6 | 32.8 | 33.0 |
| GPT-4o-05 | 29.5 | 33.2 | 39.4 | 40.8 | 43.6 |
| GPT-4o-08 | 28.3 | 34.3 | 38.9 | 40.1 | 43.2 |
| GPT-4o-mini | 26.7 | 33.8 | 35.1 | 39.0 | 40.8 |
| Claude-3.5-Sonnet | 28.9 | 33.8 | 36.5 | 35.6 | 36.6 |
| Gemini-1.5-Flash | 25.0 | 28.4 | 33.5 | 38.3 | 39.4 |
| Gemini-1.5-Pro | 30.0 | 32.7 | 36.3 | 41.8 | 46.4 |
| Llama-3.1-8B | 21.3 | 24.2 | 27.5 | 26.7 | 27.0 |
| Llama-3.1-70B | 26.1 | 31.3 | 34.2 | 32.9 | 31.6 |
| Llama-3.3-70B | 24.9 | 30.0 | 33.4 | 32.9 | 33.3 |
| Mistral-Nemo | 25.3 | 19.8 | 20.8 | 19.3 | 18.5 |
| MegaBeam-Mistral | 20.3 | 22.6 | 24.1 | 29.0 | 28.9 |
| Ministral-8B | 22.7 | 25.6 | 23.1 | 14.4 | 11.5 |
| Phi-3-mini-128k | 19.1 | 21.7 | 24.0 | 25.0 | 23.7 |
| Phi-3-small-128k | 18.6 | 20.3 | 24.8 | 24.4 | 6.6 |
| Phi-3-med-128k | 21.4 | 23.3 | 23.0 | 27.6 | 24.8 |
| Phi-3.5-mini | 19.6 | 21.1 | 22.9 | 24.4 | 26.3 |
| Qwen2.5-72B | 28.2 | 33.9 | 38.8 | 34.1 | 31.4 |
| Qwen2.5-7B-1M | 22.9 | 25.0 | 27.5 | 27.9 | 28.9 |
| Qwen2.5-14B-1M | 27.3 | 32.1 | 34.3 | 35.3 | 38.6 |
| Jamba-1.5-Mini | 17.1 | 17.3 | 17.3 | 17.7 | 18.1 |
| ProLong | 21.7 | 22.3 | 22.2 | 26.0 | 29.2 |

**ICL**

| | 8K | 16K | 32K | 64K | 128K |
|---|---|---|---|---|---|
| GPT-4 | 80.8 | 82.0 | 76.5 | 59.0 | 46.0 |
| GPT-4o-05 | 79.9 | 80.1 | 75.8 | 63.0 | 48.4 |
| GPT-4o-08 | 82.6 | 85.9 | 87.2 | 86.8 | 86.3 |
| GPT-4o-mini | 77.3 | 79.9 | 81.9 | 81.9 | 81.2 |
| Claude-3.5-Sonnet | 87.1 | 88.7 | 89.9 | 89.2 | 61.0 |
| Gemini-1.5-Flash | 74.2 | 68.7 | 56.6 | 39.7 | 28.1 |
| Gemini-1.5-Pro | 78.7 | 81.1 | 82.0 | 80.4 | 79.4 |
| Llama-3.1-8B | 71.3 | 76.4 | 77.9 | 81.4 | 83.9 |
| Llama-3.1-70B | 71.6 | 75.0 | 76.1 | 81.6 | 81.4 |
| Llama-3.3-70B | 70.2 | 75.0 | 76.3 | 82.1 | 77.8 |
| Mistral-Nemo | 68.6 | 76.0 | 80.2 | 82.2 | 84.0 |
| MegaBeam-Mistral | 72.1 | 76.4 | 80.6 | 83.8 | 86.2 |
| Ministral-8B | 71.3 | 74.8 | 77.3 | 82.8 | 84.4 |
| Phi-3-mini-128k | 65.6 | 69.6 | 73.7 | 76.2 | 78.7 |
| Phi-3-small-128k | 67.8 | 74.1 | 79.1 | 82.3 | 79.6 |
| Phi-3-med-128k | 62.6 | 65.5 | 67.8 | 73.0 | 73.2 |
| Phi-3.5-mini | 63.8 | 70.2 | 75.2 | 78.0 | 79.5 |
| Qwen2.5-72B | 78.4 | 80.4 | 81.8 | 83.6 | 83.2 |
| Qwen2.5-7B-1M | 70.8 | 69.2 | 57.2 | 49.4 | 47.1 |
| Qwen2.5-14B-1M | 79.8 | 84.7 | 88.2 | 89.6 | 90.5 |
| Jamba-1.5-Mini | 81.0 | 85.0 | 87.9 | 89.7 | 91.0 |
| ProLong | 69.1 | 75.1 | 78.9 | 83.2 | 86.5 |

**Avg.**

| | 8K | 16K | 32K | 64K | 128K |
|---|---|---|---|---|---|
| GPT-4 | 66.0 | 64.5 | 57.3 | 46.3 | 39.2 |
| GPT-4o-05 | 64.2 | 63.9 | 63.3 | 60.1 | 56.4 |
| GPT-4o-08 | 65.5 | 67.1 | 66.2 | 65.4 | 64.8 |
| GPT-4o-mini | 61.2 | 61.4 | 60.1 | 57.7 | 55.7 |
| Claude-3.5-Sonnet | 58.6 | 50.2 | 49.5 | 44.5 | 38.4 |
| Gemini-1.5-Flash | 62.1 | 62.0 | 60.2 | 56.1 | 52.1 |
| Gemini-1.5-Pro | 62.9 | 63.7 | 65.4 | 64.0 | 64.4 |
| Llama-3.1-8B | 56.0 | 54.3 | 52.3 | 52.0 | 46.5 |
| Llama-3.1-70B | 61.5 | 61.8 | 61.2 | 58.2 | 49.7 |
| Llama-3.3-70B | 61.3 | 62.1 | 61.4 | 58.3 | 48.2 |
| Mistral-Nemo | 55.9 | 49.2 | 38.4 | 28.7 | 25.7 |
| MegaBeam-Mistral | 50.6 | 49.2 | 48.4 | 47.4 | 45.4 |
| Ministral-8B | 56.1 | 53.0 | 47.7 | 28.5 | 24.2 |
| Phi-3-mini-128k | 49.9 | 47.5 | 45.5 | 43.3 | 33.6 |
| Phi-3-small-128k | 48.1 | 47.2 | 45.7 | 43.2 | 24.9 |
| Phi-3-med-128k | 51.0 | 46.5 | 41.0 | 37.7 | 29.1 |
| Phi-3.5-mini | 48.8 | 46.9 | 44.6 | 42.0 | 33.6 |
| Qwen2.5-72B | 63.4 | 63.3 | 61.8 | 51.4 | 38.2 |
| Qwen2.5-7B-1M | 56.4 | 53.3 | 48.0 | 44.6 | 41.3 |
| Qwen2.5-14B-1M | 60.9 | 59.9 | 59.0 | 57.2 | 56.0 |
| Jamba-1.5-Mini | 53.4 | 50.9 | 50.6 | 49.3 | 46.9 |
| ProLong | 54.4 | 53.7 | 51.1 | 50.3 | 49.4 |

Figure 6: Results of HELMET. All models are instruction-tuned and have a claimed context window of 128K tokens or more.

**Performance degradation with longer inputs is category-dependent.** Most frontier models largely retain performance on recall and RAG with longer inputs; however, even the best models experience significant degradation as context length increases on tasks like re-ranking and generation with citations. As illustrated in Figure 7, performance degradation at longer lengths becomes more pronounced as task complexity increases from left to right. On generation with citations, open-source models completely collapse at 128K, while GPT-4o remains relatively stable. This underscores the importance of evaluating models on more complex long-context applications.

**No clear winner across all categories.** As we observe from the previous sections, the different categories do not always correlate with each other. This is evident in the varying top-performing models across categories: for instance, GPT-4o excels in recall and generation with citations, while Gemini performs better in passage re-ranking and long-document QA. Furthermore, many open-source models outperform closed-source models on ICL, potentially because heavy instruction tuning negatively impacts ICL. We provide qualitative examples in Table 19. Thus, evaluating models across multiple axes is essential. In the appendix, we also present additional analysis, such as the performance of positional extrapolation methods (§E.3), the lost-in-the-middle phenomenon (§E.4), and the performance of Claude (§E.6).

| | NIAH | | | | | RAG | | | | | Re-rank | | | | | Cite | | | | |
|---|---|---|---|---|---|---|---|---|---|---|---|---|---|---|---|---|---|---|---|---|
| GPT-4o-08 | 100.0 | 100.0 | 100.0 | 100.0 | 100.0 | 73.9 | 74.0 | 72.4 | 71.5 | 70.2 | 86.9 | 80.5 | 68.5 | 58.4 | 50.0 | 45.4 | 46.7 | 45.4 | 45.4 | 44.3 |
| Gemini-1.5-Pro | 96.8 | 98.0 | 96.0 | 34.0 | 36.2 | 73.6 | 73.4 | 72.3 | 71.8 | 71.1 | 87.6 | 84.1 | 76.5 | 68.4 | 59.7 | 47.2 | 45.9 | 44.8 | 46.2 | 43.6 |
| Llama-3.1-8B-Inst | 100.0 | 100.0 | 100.0 | 100.0 | 100.0 | 70.8 | 68.8 | 66.5 | 65.8 | 59.5 | 67.0 | 51.7 | 43.4 | 32.8 | 14.0 | 36.3 | 26.7 | 10.9 | 13.1 | 2.9 |
| Llama-3.1-70B-Inst | 100.0 | 100.0 | 100.0 | 100.0 | 100.0 | 74.0 | 71.5 | 70.9 | 69.5 | 56.2 | 83.5 | 73.5 | 60.2 | 41.0 | 24.5 | 44.3 | 41.3 | 40.1 | 29.8 | 7.5 |
| Jamba-1.5-Mini | 100.0 | 100.0 | 100.0 | 100.0 | 100.0 | 65.2 | 64.1 | 63.4 | 61.8 | 57.3 | 62.1 | 46.3 | 33.4 | 25.4 | 14.6 | 14.4 | 8.3 | 6.6 | 6.4 | 3.1 |
| Qwen2-7B-Inst | 100.0 | 100.0 | 100.0 | 100.0 | 100.0 | 58.5 | 57.3 | 54.0 | 51.1 | 47.4 | 47.9 | 28.2 | 19.5 | 8.6 | 5.9 | 15.5 | 7.4 | 3.6 | 3.5 | 2.4 |
| Qwen2-57B-Inst | 100.0 | 100.0 | 100.0 | 87.0 | 36.0 | 65.4 | 66.2 | 62.2 | 54.7 | 12.7 | 53.3 | 38.9 | 22.3 | 6.5 | 0.0 | 36.0 | 11.3 | 4.1 | 3.9 | 1.1 |
| | 8K | 16K | 32K | 64K | 128K | 8K | 16K | 32K | 64K | 128K | 8K | 16K | 32K | 64K | 128K | 8K | 16K | 32K | 64K | 128K |

Figure 7: Results of selected instruct models on various lengths and increasing complexity of tasks. Notably, Qwen2 relies on position extrapolation, while other open-models are trained at or greater than 128K context window.

## 4 RELATED WORKS

**Long-context language models.** Frontier models such as GPT-4 (OpenAI, 2023), Gemini (Team et al., 2024b), and Claude claim to have expanded their context window beyond 100K tokens. In the open-source community, there are also efforts to train models with longer input lengths (Dubey et al., 2024; Fu et al., 2024; AI et al., 2024; Gao et al., 2024), explore position extrapolation techniques (Peng et al., 2024; Chen et al., 2023; Ding et al., 2024), and experiment with efficient architectures (Beltagy et al., 2020; Bertsch et al., 2023; Gu & Dao, 2024; Dao & Gu, 2024; Yen et al., 2024; Lieber et al., 2024, *inter alia*).

**Synthetic tasks.** Synthetic tasks are often used to evaluate LCLMs since they can be procedurally generated, enabling arbitrarily long input lengths and controlled "needle" placement (Tay et al., 2021; Liu et al., 2023). In particular, Needle-in-a-Haystack (NIAH; Kamradt, 2024) inserts a "needle" at specific depths of a long essay (i.e., the haystack) and asks the model to recall the fact. Recent works have expanded upon it and designed new procedures to test different aspects of LCLMs (Hsieh et al., 2024; Li et al., 2024b; Levy et al., 2024; Arora et al., 2023; Laban et al., 2024; Goldman et al., 2024, *inter alia*). However, they do not study how results on synthetic tasks transfer to real applications. In contrast, we evaluate both synthetic and downstream tasks and investigate how they correlate with each other.

**Long-context benchmarks.** As discussed in the main paper, existing benchmarks either are limit to relatively short context lengths (Shaham et al., 2022; 2023; Li et al., 2024a; An et al., 2024; Bai et al., 2024; Dong et al., 2024; Kwan et al., 2024) or lack rigorous evaluation methods (Zhang et al., 2024b; Yuan et al., 2024). Many works instead focus on specific domains, such as question answering (Wang et al., 2024b; Karpinska et al., 2024; Wang et al., 2024a;c), in-context learning (Li et al., 2024c; Bertsch et al., 2024; Xu et al., 2024; Anil et al., 2024; Agarwal et al., 2024), summarization (Chang et al., 2024; Kim et al., 2024; Shen et al., 2022), or RAG (Lee et al., 2024). In this work, we construct a comprehensive benchmark that tests across diverse downstream tasks at long input lengths and also present a unified comparison and analysis across 59 LCLMs.

## 5 CONCLUSION

In this work, we first identify the shortcomings of long-context evaluation settings and existing benchmarks—over-reliance on synthetic tasks, limited coverage of realistic applications, and unreliable metrics among others. We seek to address these issues by constructing HELMET, an application-centric benchmark with diverse domains and reliable evaluation settings. We then present a comprehensive evaluation of 59 frontier LCLMs across multiple dimensions, including different tasks, input lengths, and model types. Our analysis shows that synthetic tasks poorly predict downstream performance and that different categories exhibit distinct capabilities and trends. Thus, evaluating models on a diverse set of real-world tasks is essential. Furthermore, open-source models still lag behind closed-source models on complex tasks at longer lengths. Finally, we hope that our benchmark and comprehensive evaluation serve as a valuable resource for future research in long-context language modeling.

## REPRODUCIBILITY STATEMENT

We have publicly released all of our code and data to ensure reproducibility. Experimental settings are outlined in the main text (§3) and the appendix (§D), and we will provide additional details and instructions in the code repository. Furthermore, we plan on releasing the model generated outputs and statistics to enable further analysis and comparison.

## ACKNOWLEDGMENTS

We would like to acknowledge Mengzhou Xia, Howard Chen, Xi Ye, Yinghui He, Lucy He, Alexander Wettig, Sadhika Malladi, and other members of Princeton Language and Intelligence for their helpful feedback and discussion. We also thank Adithya Bhaskar and Joie Zhang for helping with validating the model-based evaluation. Tianyu Gao is supported by an IBM PhD Fellowship. This work is gratefully supported by an NSF CAREER award (IIS-2239290), a grant from Intel, and the Microsoft Accelerate Foundation Models Research (AFMR) grant program.

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

# A  COMPARISON WITH OTHER BENCHMARKS

Due to the lack of large-scale human studies on long-context language models, discerning the true rankings of LCLMs remains challenging. Therefore, we rely on previous works and empirical observations to guide our expectations. Specifically, the Gemini 1.5 report demonstrates that larger models consistently outperform smaller ones at handling long contexts—both qualitatively and quantitatively across the Gemini, GPT, and Claude model families (Section 5.2, Tables 4 and 5 Team et al., 2024b). Although their evaluation suite is not publicly available, these results align with commonly observed patterns in language model performance. Given that obtaining ground truth rankings through human evaluation is prohibitively expensive and practically infeasible, relying on such well-established patterns is reasonable. In this section, we further examine HELMET's improvements over existing benchmarks through direct comparisons of model rankings and performance, supported by careful ablation studies.

## A.1  RESULTS

We build upon previous datasets and refine evaluation settings to better reflect model capabilities. Specifically, we evaluate models on ∞BENCH, another benchmark that evaluates long-context at 128K tokens. The results reproduced from the original authors' code are shown in Table 6. The numerical results for Figure 1 are presented in Table 5.

Upon closer inspection, we find that a set of open-sourced models' ranking and performance on HELMET and ∞BENCH in Table 9. In addition to the previously noted performance discrepancy between Llama-3.1-8B-Instruct and Llama-3.1-70B-Instruct, we observe that all Llama-3.2 models degenerate in performance on ∞BENCH at input length 128K. However, on HELMET, the Llama-3.2 models, especially Llama-3.2-3B-Instruct, rank well against other open-source models. Upon qualitative examination, we find that the Llama 3.2 models are able to produce coherent and useful generation at long contexts with better prompting strategies from HELMET, such as adding in-context learning examples.

Our ablation study on few-shot demonstrations in Table 8 shows that 2-shot demonstrations substantially improve base model performance, better reflecting real-world usage patterns. As shown in Table 7, improved instructions and prompting enable smaller Llama-3.2 models to compete effectively with larger models. Thus, HELMET provides a better and more realistic reflection of how these models would be used in practice over previous benchmarks. These findings demonstrate that HELMET provides a more accurate and practical assessment of model capabilities compared to previous benchmarks.

Table 5: Results with 128k input length across different benchmarks.

| Model | Claimed Length | NIAH | RULER | ∞BENCH | HELMET |
|---|---|---|---|---|---|
| GPT-4o-mini | 128000 | 100.0 | 80.8 | 51.9 | 54.1 |
| GPT-4o-08 | 128000 | 100.0 | 93.3 | 57.1 | 63.8 |
| Gemini-1.5-Flash | 1048576 | 100.0 | 86.6 | 50.8 | 50.7 |
| Gemini-1.5-Pro | 2097152 | 45.3 | 65.3 | 60.3 | 62.7 |
| Llama-3.1-8B-Inst | 131072 | 100.0 | 81.3 | 44.1 | 47.0 |
| Llama-3.1-70B-Inst | 131072 | 100.0 | 75.8 | 39.7 | 49.3 |

## A.2  DISCUSSIONS

Many existing long-context language modeling benchmarks are studied in isolated settings, such as synthetic tasks (Hsieh et al., 2024; Li et al., 2024b; Levy et al., 2024; Laban et al., 2024), in-context learning (Anil et al., 2024; Bertsch et al., 2024; Li et al., 2024c; Agarwal et al., 2024), summarization (Chang et al., 2024; Kim et al., 2024), and retrieval-augmented settings (Lee et al., 2024). However, these works lack a unified evaluation across diverse downstream tasks.

There are also benchmarks that sought to unify different datasets, such as ∞BENCH (Zhang et al., 2024b), LongBench (Bai et al., 2024), L-Eval (An et al., 2024), LV-Eval (Yuan et al., 2024), and

Table 6: Results on ∞BENCH(Zhang et al., 2024b). We report numbers from running the original authors repo: https://github.com/OpenBMB/InfiniteBench. The original code did not support evaluation of the Gemini and Llama 3 models at the time of access, so we evaluate them by following the script and template for GPT-4 and open-source models, respectively. We exclude coding tasks from the evaluation suite since it is out of the scope for general long-context language modeling tasks.

| Model | MC | QA | Sum | Diag | Calc | Find | Number | PassKey | KV | Avg. |
|-------|-----|-----|-----|------|------|------|--------|---------|-----|------|
| GPT4 | 75.9 | 23.7 | 10.7 | 18.5 | 0.0 | 47.1 | 100.0 | 100.0 | 58.8 | 44.9 |
| GPT-4o-05 | 88.2 | 37.9 | 23.7 | 28.5 | 0.0 | 58.6 | 100.0 | 100.0 | 94.2 | 55.4 |
| GPT-4o-08 | 86.5 | 26.0 | 21.5 | 51.0 | 0.0 | 58.9 | 100.0 | 100.0 | 99.8 | 57.1 |
| GPT-4o-mini | 78.2 | 19.1 | 24.8 | 21.5 | 0.0 | 69.7 | 100.0 | 100.0 | 80.4 | 51.9 |
| Gemini-1.5-Flash | 76.0 | 42.1 | 30.0 | 55.8 | 0.0 | 47.4 | 100.0 | 100.0 | 31.4 | 50.8 |
| Gemini-1.5-Pro | 77.5 | 27.7 | 29.0 | 97.5 | 0.0 | 58.0 | 100.0 | 100.0 | 70.4 | 60.3 |
| Llama-3.1-8B | 56.8 | 8.8 | 14.3 | 0.5 | 0.0 | 22.0 | 99.7 | 100.0 | 18.8 | 33.0 |
| Llama-3.1-8B-Inst | 67.2 | 15.5 | 26.7 | 23.0 | 0.0 | 33.1 | 99.5 | 100.0 | 55.0 | 44.1 |
| Llama-3.1-70B | 66.4 | 9.2 | 17.5 | 8.5 | 0.0 | 32.3 | 100.0 | 100.0 | 13.4 | 35.1 |
| Llama-3.1-70B-Inst | 75.5 | 23.3 | 31.3 | 18.0 | 0.0 | 43.1 | 99.7 | 100.0 | 2.6 | 39.7 |
| Llama-3.2-1B | 2.2 | 1.4 | 8.6 | 4.5 | 0.0 | 0.0 | 1.5 | 0.0 | 0.0 | 2.0 |
| Llama-3.2-1B-Instruct | 3.5 | 1.4 | 12.5 | 5.5 | 0.0 | 0.0 | 0.0 | 0.0 | 0.0 | 2.4 |
| Llama-3.2-3B | 1.3 | 1.2 | 7.6 | 4.5 | 0.0 | 0.0 | 1.7 | 0.0 | 0.0 | 1.9 |
| Llama-3.2-3B-Instruct | 2.2 | 1.6 | 13.3 | 4.5 | 0.0 | 0.0 | 1.7 | 1.7 | 0.0 | 2.8 |

Table 7: We compare the results on ∞BENCH multiple-choice (MC) and question answering (QA) tasks between the original authors and HELMET. In our implementation, we leverage refined prompts and carefully crafted parsing scripts to ensure robust and fair evaluation. For example, we find that Llama-3.2 models are actually much better long-context language models than ∞BENCH would suggest.

| Model | Claimed Length | Original | | HELMET | |
|-------|----------------|----------|-----|--------|-----|
| | | MC | QA | MC | QA |
| GPT-4o-mini | 128000 | 88.2 | 37.9 | 71.0 | 44.5 |
| GPT-4o-05 | 128000 | 86.5 | 26.0 | 76.0 | 53.0 |
| GPT-4o-08 | 128000 | 78.2 | 19.1 | 73.0 | 47.9 |
| Gemini-1.5-Flash | 1048576 | 76.0 | 42.1 | 79.1 | 51.7 |
| Gemini-1.5-Pro | 2097152 | 77.5 | 27.7 | 83.3 | 50.9 |
| Llama-3.1-8B | 131072 | 56.8 | 8.8 | 54.0 | 38.8 |
| Llama-3.1-8B-Inst | 131072 | 67.2 | 15.5 | 49.0 | 40.3 |
| Llama-3.2-1B | 131072 | 2.2 | 1.4 | 21.0 | 17.6 |
| Llama-3.2-1B-Inst | 131072 | 3.5 | 1.4 | 16.0 | 13.3 |
| Llama-3.2-3B | 131072 | 1.3 | 1.2 | 36.0 | 31.0 |
| Llama-3.2-3B-Inst | 131072 | 2.2 | 1.6 | 42.0 | 20.6 |

ZeroSCROLLS (Shaham et al., 2023). Many of these benchmarks are still limited by the context length, evaluation metrics, or both. Most similar to our work, ∞BENCHZhang et al. (2024b) also evaluates models at context lengths at 128K tokens. However, their evaluation settings are limited to a few domains—synthetic, QA, and summarization. Although they also evaluate coding tasks, the domain is limited to code-specialized models outside of the realm of general-purposed language models. Furthermore, their summarization evaluations lack robust evaluation and still rely on ROUGE scores. Similarly, LV-Eval (Yuan et al., 2024) evaluates long-context models across different lengths but is limited to the QA tasks. Furthermore, we are the first to evaluate existing LCLMs comprehensively—we evaluate 59 models of different sizes and architectures, which enables previously unavailable insights into the correlation across diverse tasks and the landscape of current models.

Table 8: We evaluate the performance of models on a subset of HELMET tasks with 0-shot and 2-shot demonstrations at 128k input length to understand the impact of ICL on model performance, averaged across three random seeds. The standard deviation across three runs are shown in the subscript. We observe that the performance is generally higher for 2-shot demonstrations compared to 0-shot demonstrations. Crucially, the 2-shot examples enable base models to achieve higher results that reflect the model's long-context capabilities in realistic settings, such as for MSMARCO and JSON KV. $^{\flat}$denotes base models.

| Model | JSON KV | | NQ | | MSMARCO | | ∞BENCH MC | | ∞BENCH QA | |
|---|---|---|---|---|---|---|---|---|---|---|
| | 0 shot | 2 shot | 0 shot | 2 shot | 0 shot | 2 shot | 0 shot | 2 shot | 0 shot | 2 shot |
| GPT-4o-mini | $92.3_{5.5}$ | $93.7_{2.1}$ | $57.8_{1.1}$ | $60.4_{2.6}$ | $24.8_{0.7}$ | $31.4_{1.1}$ | $72.0_{2.6}$ | $69.3_{4.7}$ | $28.8_{1.9}$ | $47.1_{4.4}$ |
| GPT-4o-05 | $99.3_{0.6}$ | $36.7_{7.1}$ | $59.4_{4.0}$ | $63.0_{2.1}$ | $42.7_{0.9}$ | $46.8_{1.3}$ | $73.7_{0.6}$ | $76.3_{0.6}$ | $49.1_{1.9}$ | $51.3_{3.4}$ |
| GPT-4o-08 | $100.0_{0.0}$ | $100.0_{0.0}$ | $58.5_{2.0}$ | $61.9_{4.5}$ | $46.6_{0.3}$ | $48.7_{1.0}$ | $75.3_{2.5}$ | $72.7_{0.6}$ | $29.8_{0.9}$ | $48.6_{1.5}$ |
| Gemini-1.5-Flash | $98.3_{0.6}$ | $99.0_{1.0}$ | $53.7_{5.9}$ | $51.5_{4.5}$ | $44.9_{3.0}$ | $49.9_{0.9}$ | $73.9_{2.3}$ | $77.1_{3.2}$ | $48.2_{4.7}$ | $49.9_{2.6}$ |
| Gemini-1.5-Pro | $97.7_{0.6}$ | $93.7_{2.3}$ | $62.2_{2.6}$ | $59.2_{2.7}$ | $57.1_{0.9}$ | $58.4_{0.3}$ | $83.6_{2.8}$ | $83.3_{0.7}$ | $46.4_{1.8}$ | $47.2_{3.2}$ |
| Llama-3.1-8B$^{\flat}$ | $77.3_{2.1}$ | $98.0_{1.0}$ | $45.7_{2.1}$ | $44.9_{3.9}$ | $0.1_{0.0}$ | $7.5_{0.5}$ | $55.3_{5.5}$ | $53.3_{8.0}$ | $36.1_{2.0}$ | $36.6_{2.1}$ |
| Llama-3.1-8B-Inst | $96.0_{1.0}$ | $95.7_{0.6}$ | $48.4_{1.3}$ | $48.9_{2.6}$ | $5.4_{0.0}$ | $13.8_{1.1}$ | $49.7_{2.1}$ | $52.0_{2.6}$ | $29.1_{2.0}$ | $37.9_{2.6}$ |
| Llama-3.2-1B$^{\flat}$ | $34.0_{3.5}$ | $34.3_{2.1}$ | $25.1_{0.7}$ | $23.2_{2.8}$ | $0.3_{0.0}$ | $4.8_{0.5}$ | $23.0_{2.6}$ | $22.7_{1.5}$ | $18.9_{2.5}$ | $18.9_{3.3}$ |
| Llama-3.2-1B-Inst | $6.3_{1.5}$ | $9.3_{3.2}$ | $23.3_{4.4}$ | $22.9_{1.4}$ | $0.0_{0.0}$ | $0.5_{0.3}$ | $16.0_{2.6}$ | $16.3_{2.5}$ | $12.6_{0.4}$ | $12.7_{0.5}$ |
| Llama-3.2-3B$^{\flat}$ | $30.3_{2.1}$ | $54.0_{8.9}$ | $40.2_{2.0}$ | $42.3_{4.1}$ | $0.1_{0.0}$ | $6.0_{1.3}$ | $43.0_{1.0}$ | $40.7_{7.2}$ | $31.3_{1.1}$ | $31.6_{2.2}$ |
| Llama-3.2-3B-Inst | $31.3_{2.3}$ | $36.7_{2.5}$ | $44.2_{2.5}$ | $42.8_{3.8}$ | $0.2_{0.0}$ | $0.9_{0.2}$ | $36.0_{2.0}$ | $40.0_{3.5}$ | $17.8_{1.2}$ | $19.7_{0.8}$ |

Table 9: Comparison of model rankings on HELMET and ∞BENCH.

| Model | HELMET | Model | InfBench |
|---|---|---|---|
| Llama-3.1-70B-Inst | 49.3 | Llama-3.1-8B-Inst | 46.7 |
| Llama-3.1-8B-Inst | 47.0 | Llama-3.1-70B-Inst | 43.7 |
| Llama-3.1-70B | 41.3 | Yi-34B-200k | 43.1 |
| Yi-34B-200k | 38.3 | Llama-3.1-70B | 38.6 |
| Llama-3.2-3B-Inst | 36.9 | Yi-9B-200k | 37.6 |
| Llama-3.1-8B | 35.6 | Llama-3.1-8B | 35.7 |
| Yi-9B-200k | 33.0 | Yi-6B-200k | 32.0 |
| Llama-3.2-3B | 31.9 | Llama-3.2-3B-Inst | 2.8 |
| Yi-6B-200k | 26.3 | Llama-3.2-1B-Inst | 2.6 |
| Llama-3.2-1B-Inst | 24.6 | Llama-3.2-1B | 2.0 |
| Llama-3.2-1B | 21.2 | Llama-3.2-3B | 1.8 |

To summarize, HELMET improves upon existing benchmarks by providing a comprehensive evaluation across diverse tasks and carefully designed evelation metrics and prompting strategies. Consequently, HELMET provides a more accurate reflection of the models' capabilities in practice.

# B DATASETS

## B.1 RETRIEVAL-AUGMENTED GENERATION

Natural Questions (NQ; Kwiatkowski et al., 2019) is a large-scale dataset for open-domain question answering featuring real user queries. TriviaQA (TQA; Joshi et al., 2017) comprises trivia questions and their corresponding answers. HotpotQA (Yang et al., 2018) contains questions that require multi-passage reasoning. We source these datasets from KILT (Petroni et al., 2021), which provides annotations linking each query to its corresponding gold passages containing the answers.

PopQA (Mallen et al., 2023) consists of questions about long-tail entities. To minimize the impact of pre-training memorization, we filter the dataset to include only subject entities with popularity scores below 1000. Since gold passages are not available for PopQA, we classify retrieved passages as positive or negative by checking for the presence of the ground truth answer.

Notably, we populate the context with *hard negative passages*, retrieved from the same corpus as the positive passages using a real retriever. This approach presents a significantly greater challenge than using randomly sampled passages. This design choice better reflects real-world retrieval-augmented

generation tasks, where models must effectively distinguish between relevant and irrelevant information.

## B.2 GENERATION WITH CITATIONS

Generation with citations represents a crucial task for enhancing language model trustworthiness and verifiability (Bohnet et al., 2022; Gao et al., 2023; Asai et al., 2024b). We employ ALCE to assess models' capability to generate properly cited text. Following the original methodology, we utilize Wikipedia as the retrieval corpus and GTR (Ni et al., 2022) as the retriever. We omit MAUVE-based fluency evaluation since models typically generate fluent text, and in cases where they don't, the other metrics already approach zero. Instead, we focus on correctness and citation quality, reporting their average.

## B.3 PASSAGE RE-RANKING

We evaluate models' passage re-ranking capabilities (Sun et al., 2023) using the MS MARCO passage ranking dataset (Bajaj et al., 2018). Our evaluation uses annotated datasets from the TREC Passage Re-ranking challenge (Craswell et al., 2020). Each dataset instance consists of a query and a list of passages with associated relevance scores. The dataset combines Bing user queries with passages retrieved via BM25 from web pages. Passages are labeled as perfect, highly relevant, or not relevant. For each input length $L$, we determine the number of passages $k$ that can be included. To eliminate positional bias, we balance the label distribution and randomize passage order. We create three different permutations of the $k$ passages for each input. The final performance is reported in NDCG@10. While higher values of $k$ are possible for NDCG evaluation, the computational cost of generating numerous passage IDs during inference leads us to maintain NDCG@10 as our metric.

## B.4 IN-CONTEXT LEARNING

For in-context learning, we implement a label mapping strategy that compels models to utilize in-context examples rather than relying on pre-trained knowledge for classification tasks. Each label is randomly mapped to an integer $l \in \{0, 1, \ldots, n-1\}$, where $n$ represents the number of unique labels in the dataset, following established practices (Pan et al., 2023). We then replace all label texts with their corresponding integer mappings throughout the dataset. Following Li et al. (2024c), we organize examples into demonstration rounds, with each round containing exactly one example per label in randomized order. The input is constructed by concatenating these demonstration rounds until reaching the target input length $L$. Unlike other task categories where we evaluate 100 samples, for ICL datasets we evaluate 500 samples. We balance the label distribution among the evaluation set.

## B.5 SYNTHETIC TASKS

For RULER tasks, we generate data using the original authors' scripts (Hsieh et al., 2024), employing the Llama-2 tokenizer (Touvron et al., 2023) to standardize input text across all models. The RULER suite comprises the following tasks:

- NIAH Single (three variants):
    - NIAH (variant 2): Most similar to the original NIAH (Kamradt, 2024)
    - NIAH Single Repeat (variant 1): Uses repeated phrases instead of Paul Graham essays as context
    - NIAH Single UUID (variant 3): Similar to variant 2 but uses UUIDs as retrieval values
- NIAH MultiKey (MK, three variants):
    - NIAH MK Essay (variant 1): Involves retrieving one gold key among three irrelevant keys, using Paul Graham Essays as context
    - NIAH MK Needle (variant 2): Uses needle-based context structure
    - NIAH MK UUID (variant 3): Similar to variant 2 but employs UUIDs as retrieval values for all needles

- NIAH MultiValue (MV): Requires retrieving four different numbers associated with a single key from irrelevant essay context
- NIAH MultiQuery (MQ): Involves retrieving correct values for four distinct keys from irrelevant essay context
- Variable Tracking (VT): Requires tracking variable values through sequential operations
- Word Extraction Tasks:
  - Common word extraction (CWE)
  - Frequent word extraction (FWE)
  - Both tasks require identifying the most frequently occurring words
- Question Answering Tasks:
  - SQuAD (variant 1; Rajpurkar et al., 2016)
  - HotpotQA (variant 2; Yang et al., 2018)
  - Notable distinction: Our implementation uses retrieved passages rather than random samples, making the task more realistic and challenging

For comprehensive details, please refer to the original paper (Hsieh et al., 2024).

For JSON KV tasks, we generate a JSON dictionary containing randomly generated UUIDs as both keys and values, similar to Liu et al. (2023). For each dictionary, we construct six queries, asking the model to retrieve the value for each key at six evenly spaced positions within the context.

### B.6    MODEL-BASED EVALUATION

Automatic evaluation metrics, such as ROUGE, are known to be unreliable and uncorrelated with human judgments (Goyal et al., 2023; Chang et al., 2024). However, existing long-context benchmarks still largely rely on these metrics (Bai et al., 2024; Zhang et al., 2024b). In this work, we seek to more reliably evaluate language models at long-contexts by leveraging LLMs as judges, inspired by previous works (Zheng et al., 2023). For all model-based evaluations, we use `GPT-4o-2024-05-13` as the judge.

**Long-document question answering.** For NarrativeQA, we found that the models can often output answers that are correct but have little surface form or lexical overlaps with the ground truth answers in preliminary experiments, resulting in lower-than-expected performance. This is often due to the long lengths of the ground truth answers, which gives more possibilities of how to write it than a simple named entity that is often the case of other QA datasets.

Therefore, we employ an LLM to judge if the model output is fluent and semantically similar to the ground truth. Given the question, the ground truth answer, and the model-generated output, we ask the LLM to judge if (1) the model output is fluent and (2) the generated output is correct and relevant to the question with the ground truth as a reference. The detailed prompts, precise definitions, and instructions are found in Table 10.

In Table 14, we also find that model evaluation can be useful in catching extremely cases. For example, Claude scores low in terms of F1 due to its tendency to output extra, assistant-like text, which is penalized by the n-gram overlap metric, and appears to be worse than the much smaller Llama-3.2-3B-Inst model. However, the model-based evaluation is able to catch this issue, and Claude scores higher than Llama-3.2-3B-Inst, which users might have expected given the model sizes.

**Summarization**. At a high level, we ask the model to check for three properties: fluency, precision, and recall. Fluency can take on a value of either 0 (incoherent, incomplete, and/or repetitive) or 1 (fluent and coherent). Precision is the percentage of model-generated sentences that are supported by the gold summary (we use the long summary from Multi-LexSum here). Recall is the percentage of the key points that are supported by the model-generated output. We calculate the F1 score from precision and recall and multiply it with the fluency score for the final score.

We first ask the model to generate a list of key points or atomic claims from the gold summary, following previous works that show that LLMs can accurately decompose long texts (Kamoi et al.,

2023; Gao et al., 2023). We manually checked 105 claims originating from 25 Multi-LexSum summaries and found that the claims were all factually correct. Although we found one out of 25 instances where the model missed a possible key point in the given summary, we found that GPT-4o is almost always reliable for the decomposition task. For Multi-LexSum, we use the short summary to obtain the key point as the annotation contains both a long and a short summary. These key points are then saved for Multi-LexSum and ∞BENCH Sum.

We show the detailed prompts for the summarization tasks in Table 11, 12, and 13, which are modeled after previous works (Kamoi et al., 2023; Chang et al., 2024; Kim et al., 2024). These previous works have shown that LLMs can effectively judge model outputs (Zheng et al., 2023), but we conduct human analysis to further verify the evaluation metric.

From qualitative analysis, we found that the model is consistently able to distinguish between fluent and non-fluent outputs, where we agree with the model judgments 100% of the time for randomly sampled Multi-LexSum and ∞BENCH Sum outputs. We then check if the GPT-4o judgments for precision and recall agree with human judgments. To this end, we sample 10 generated summaries for Multi-LexSum and ∞BENCH each (from Gemini-1.5-Pro and Llama-3.1-70B-Inst) and check five key point evaluations for each summary. We follow a similar procedure where we check if the model output supports the key point. We observed Cohen's $\kappa = 0.76$ for ∞BENCH Sum and $\kappa = 0.72$ for Multi-LexSum, suggesting substantial agreement. For precision, we conduct a similar human evaluation, and found a $\kappa = 0.91$ for ∞BENCH Sum and $\kappa = 0.83$ for Multi-LexSum, suggesting near-perfect agreement.

Inspecting the disagreements, we find that that most of them arise from the partially supported cases. For instance, the key point may include specific details, such as the names of government departments or Court Justices' names, that are not explicitly mentioned in the generated summary, and the model judge is typically more lenient about the exclusion of these small details while humans are more strict. However, this is also subjective to the preference of the human.

Qualitatively, We find that model-based evaluation can catch cases where the model is overly repetitive and scores high ROUGE-L score as a result, such as Mistral-7B-Inst-v0.3 on ∞BENCH Sum. For instance, the Mistral model may generate an output consisted of the sentence "The author's object is to show that the study of grammar is necessary part of education" repeated for hundreds of times. This summary would receive an ROUGE-L score of 12.3 while the GPT-4o judge would penalize the model for being overly repetitive and incoherent and assign the output a final score of 0.0. Thus, our GPT-4o judge penalizes the model for being overly repetitive, and the final metric reflects this issue.

## C MODELS

Please see Table 15 for the detailed information of the models used in this work.

## D EXPERIMENTAL SETUP

As previously described, we are able to evaluate models across different input lengths. Thus, we evaluate all models at $L \in \{8192, 16384, 32768, 65536, 131072\}$. For the proprietary models, GPT, Gemini, and Claude, we rely on the provider's API. For all open-source models, we evaluate on a H100 GPUs with 80GB of memory. We use the HuggingFace framework (Wolf et al., 2020) to load and generate model outputs. We apply instruction-tuned models' chat template whenever applicable. We use FlashAttention2 (Dao, 2023) and BF16 for faster inference. Our compute is limited to 8 H100 GPUs; thus, we are not able to run some of the larger models, such as Command-R or Jamba-1.5-Large, at 128K tokens. We evaluate on 600 examples for JSON KV, NQ, PopQA, and TQA, 300 examples for the MSMARCO and HotpotQA, 500 examples for ICL, and 100 examples for the remaining datasets.

Table 10: Model-based evaluation prompt for long-document question answering.

```
Please act as an impartial judge and evaluate the quality of the
provided answer which attempts to answer the provided question based on
a provided context.  Although you are not given the context, you will be
given a set of correct answers that achieves full scores on all metrics,
and you need to assess the provided answers using the correct answers.

Below is your grading rubric:

Fluency:  - Score 0 (incoherent, repetitive, or incomplete):  Incoherent
sentences, repetitive sentences (even if not by exact words), incomplete
answers, or gibberish.  Note that even if the answer is coherent, if it
is repetitive or incomplete, it should be given a score of 0.  - Score 1
(coherent, non-repetitive answer):  Coherent, non-repetitive, fluent,
grammatically correct answers.

Correctness:  - Score 0 (Incorrect):  The answer does not agree with the
provided correct answers at all.  - Score 1 (partly correct):  Partly
agree with one of the provided correct answers (for example, the
question asks for a date and a person; the answer gets the date right
but the person wrong).  - Score 2 (correct but not fully relevant):
Fully agrees with one of the provided correct answers but mentions other
completely irrelevant information.  Note that extra details provided in
the answer, even if not mentioned in the correct answers, should NOT be
seen as irrelevant as long as they are relevant to the question to a
reasonable extend.  - Score 3 (correct and relevant):  Fully agrees with
one of the provided correct answers and only provides information
relevant to the question.  Note that if the answer is longer than the
correct answer, as long as everything in the answer is relevant to the
question, it should still be given score 3.  For example, if the correct
answer is ''the North Pole'' and the answer is ''They are headed for the
North Pole'', it should still be given a score of 3.

Now, read the following question, answer, and correct answers.  First
think step-by-step and provide your reasoning and assessment on the
answer.  Then output your score in the following json format:
{{''fluency'':  0, ''correctness'':  1}}.

Question:  {question}
Correct answers:  {correct_answers}
Answer:  {parsed_output}
```

# E  ADDITIONAL RESULTS

## E.1  CORRELATION BETWEEN SYNTHETIC AND DOWNSTREAM TASKS

We show the correlation between all synthetic and RAG datasets with other downstream tasks in Figure 8. We see that the synthetic tasks are generally less correlated with the downstream tasks, whereas the RAG datasets are more correlated with the downstream tasks. Furthermore, there appears to be a pattern between the complexity of the synthetic task and its correlation with other tasks—the more complex the task, the more correlated it is with the downstream tasks. Furthermore, noisier variants tend to be more reflective of real-world applications, as seen in the difference between RULER MK Essay, where the context is irrelevant essays, and RULER MK Needle, where the context is other distracting needles. Upon closer inspection, we note that Needle MK Needle/UUID, JSON KV, and NIAH MV generally have relatively high correlation with other realistic datasets, and are able to test the recall abilities of the model. Thus, we select these four datasets as part of the HELMET synthetic recall subset. We validate this selection by checking the correlation

Table 11: Model-based evaluation prompt for summarization fluency score.

```
Please act as an impartial judge and evaluate the fluency of the
provided text.  The text should be coherent, non-repetitive, fluent, and
grammatically correct.

Below is your grading rubric:  - Score 0 (incoherent, repetitive, or
incomplete):  Incoherent sentences, repetitive sentences (even if not by
exact words), incomplete answers, or gibberish.  Note that even if the
answer is coherent, if it is repetitive or incomplete, it should be
given a score of 0.  - Examples:  - Incomplete:  ''Summary:''  -
Incoherent:  ''Summary:  The plaintiff the the the the able the the the
the the the the the the the the able the the the the the ''  - Repetitive:
''Summary:  The U.S. government brought a criminal case against four
defendants.  Summary:  The U.S. government brought a criminal case
against four defendants.  Summary:  The U.S. government brought a
criminal case against four defendants.  Summary:  The U.S. government
brought a criminal case against four defendants.''

- Score 1 (coherent, non-repetitive answer):  Coherent, non-repetitive,
fluent, grammatically correct answers.  If the text is coherent,
non-repetitive, and fluent, but the last sentence is truncated, it
should still be given a score of 1.  - Examples:  - ''This case is about
an apprenticeship test that had a disparate impact on Black
apprenticeship applicants.  The Equal Employment Opportunity Commission
(EEOC) filed this lawsuit on December 27, 2004, in U.S. District Court
for the Southern District of Ohio.''  - ''The plaintiffs sought
declaratory and injunctive relief, as well as attorneys' fees and costs,
under the Americans with Disabilities Act, the Rehabilitation Act of
1973, the Social Security Act, and the Nursing Home Reform Act.  The
case was certified as a class action on behalf of all Medicaid-eligible
adults with disabilities in Cook County, Illinois, who are being, or may
in the future be, unnecessarily confined to nursing facilities and with
appropriate supports and services may be able to live in a community
setting.  The defendants denied the allegations and argued that the
plaintiffs' claims were not typical of the class and that the class
definition was too broad.  The case is ongoing, with discovery and
expert testimony scheduled for the fall of''

Now, read the provided text, and evaluate the fluency using the rubric.
Then output your score in the following json format: {{''fluency'':
1}}.

Text:  ''{text}''
```

between this set and other downstream tasks, and found that it generally has higher correlation than using synthetic datasets individually. There may be more optimal methods for selecting synthetic datasets, and we leave this as future work.

### E.2 CORRELATION BETWEEN DATASETS

We plot the correlation between all HELMET datasets in Figure 9. In general, the datasets in each category are strongly correlated with each other. One exception to this observation is the lack of correlation between the ALCE datasets. This suggests that writing the correct citations is a different skill from answering questions with facts. We also observe some low correlation between datasets for long-document QA and ICl, but this is likely due to the diversity within each categories.

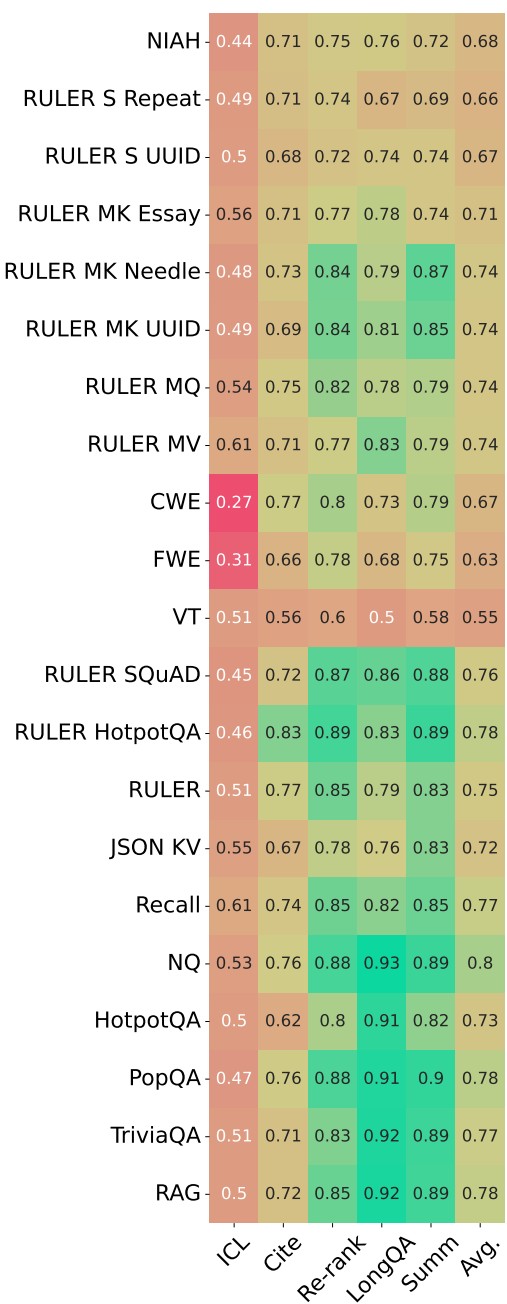

Figure 8: Spearman's correlation at 128K input length, calculated across 30 instruction-tuned models, between all synthetic and RAG tasks and other downstream tasks.

### E.3 POSITIONAL EMBEDDING EXTRAPOLATION REMAINS A CHALLENGE

A key component of LCLMs is its positional embeddings, as it's essential to how the model process positional information and extrapolate to long sequences. Thus, we also consider models that leverages positional extrapolation during inference. Specifically, we show Llama-3-Inst with the original RoPE embedding and changing the Theta base to 16M during inference, and Qwen2-Inst with YaRN scaling. Formally, RoPE defines $\theta_d = b^{-2d/|D|}$, where $\theta_d$ is the angle at the $d$-th hidden state, $b$ is a constant called the base, and $|D|$ is the size of the hidden state (Su et al., 2021).

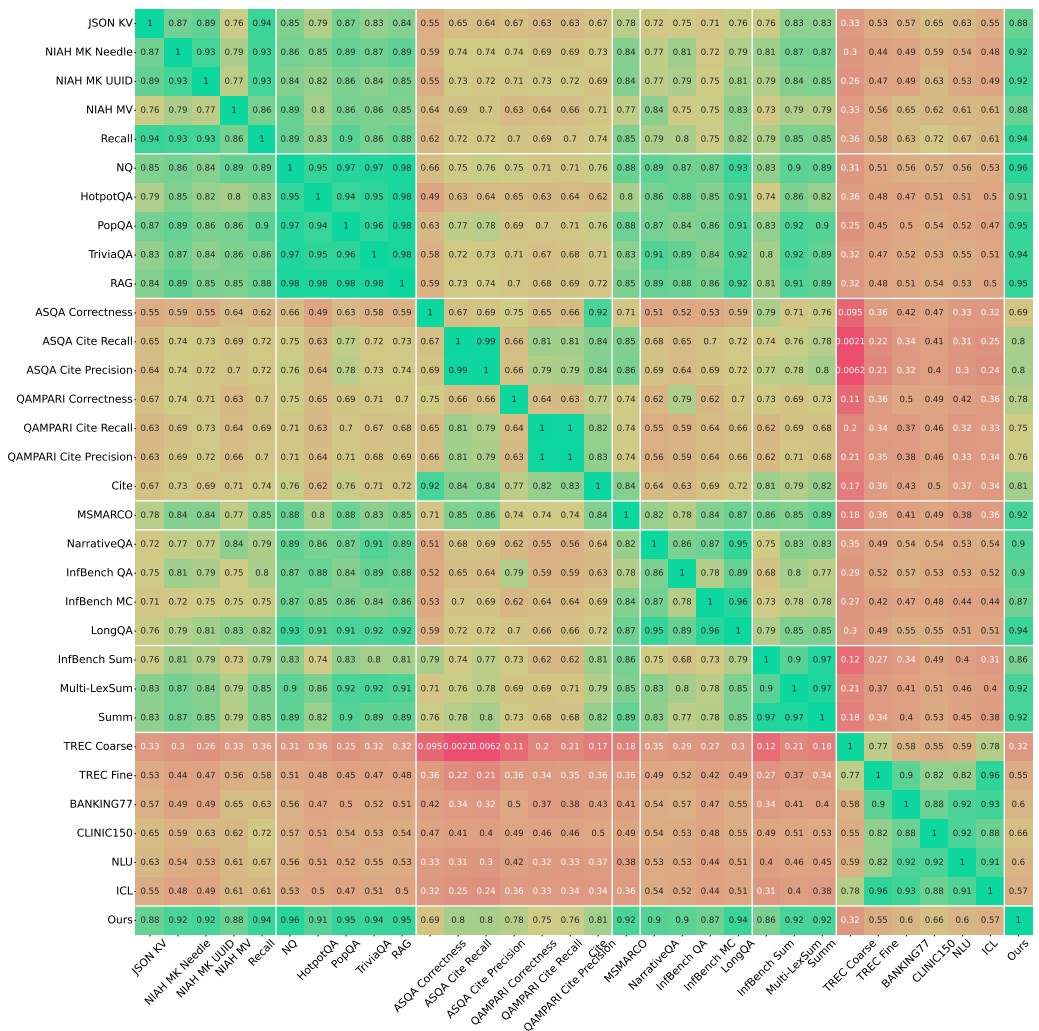

Figure 9: Spearman's correlation at 128K input length, calculated across 35 instruction-tuned models, between all HELMET datasets and category averages.

Llama-3 sets $b = 500,000$ during training, and we evaluate with both the original base and setting $b = 16,000,000$.

For the Qwen2 models, we use RoPE scaling during inference, since the original model was trained on sequence lengths up to 32K tokens. Specifically, we follow the recommended scaling factor specified in their HuggingFace model card[7].

We find that existing positional embeddings still struggle at out-of-distribution length, with both families of models dropping sharply in performance past $L = 32768$, show in Figure 7. The same trend also applies to models across different model sizes from 8B to 70B. Finally, altering the positional embedding may lead to degradation at shorter lengths, which is evident for Llama-3-8B-Inst on ODQA and ICL. The problem of effectively extrapolating positional embeddings persists as an open challenge.

### E.4 PRESENCE OF LOST IN THE MIDDLE

Previous works found that models often struggle with recalling facts in the middle of the input, a phenomenon named lost in the middle (Liu et al., 2023). In this work, we extend the previous

_______________

[7]https://huggingface.co/Qwen/Qwen2-7B-Instruct#processing-long-texts

analysis to input length up to 128K tokens. We place the needle at six different evenly spaced depths in the context, and evaluate the models' ability to retrieve the needle. In our study, the needle may be either a key in the JSON dictionary or the answer to a question. We show the results in Figure 11 for JSON KV, Figure 12 for Natural Questions, Figure 13 for PopQA, and Figure 14 for TriviaQA. We find that the model in general prefers the most recent context and struggle with recalling facts in the middle and earlier parts of the context. Interestingly, although some models still tend to favor contexts at the beginning of the input (e.g., Llama-3 and Llama-3.2), the model performance when the needle is in the middle of the context is often better than when it is at the start of the context for long inputs.

### E.5   COMPARISON BETWEEN BASE AND INSTRUCTION-TUNED MODELS

Previous benchmarks largely focus on evaluating instruction-tuned models (Shaham et al., 2023; Zhang et al., 2024b). As a result, developers of long-context base models often turn to perplexity and synthetic tasks to evaluate their models (Fu et al., 2024; Yen et al., 2024; AI et al., 2024). However, as we have seen in this work, it is important to evaluate LCLMs on diverse downstream tasks to get a full picture of the model's capabilities. Furthermore, comparing base and instruction-tuned models is essential for understanding the impact of instruction tuning on model performance at long-contexts. Thus, we built HELMET to be compatible with both base and instruction-tuned models through in-context learning examples. Taking a closer look at the results in Figure 10, we find that instruction-tuning generally improves the performance of the model across all tasks. Instruction-tuning at long contexts may yield more benefits than simply scaling up the model sizes; for example, Llama-3.1-8B-Inst outperforms Llama-3.1-70B on most tasks.

### E.6   PERFORMANCE OF CLAUDE

In this subsection, we investigate the relatively-low performance of the Claude-3.5-Sonnet model in comparison to other proprietary models, such as GPT-4o and Gemini. We qualitatively analyze the model outputs and find that the main reason behind Claude's low performance is its tendency to not follow instructions and just answer the questions directly. For example, it often does not output the citation markers for Cite (Table 17), does not output rankings in the passage re-rankings task (Table 18), and does not output the classification label for ICL (Table 18). Sometimes it even refuses to answer the question due to copyright concerns (Table 17). Thus, it has a lower score on these tasks. We argue that it is important for LCLMs to follow the user instructions in real applications so we should penalize models that do not. Sometimes, the output may be truncated due to the restriction on the maximum number of generated tokens (Table 16). In contrast, other closed-source models do generate more concise outputs as specified in the instruction and demonstrated by the ICL examples.

### E.7   FULL RESULTS

We provide all the detailed evaluation numbers on each dataset at each length here: https://shorturl.at/5wjVl.

Table 12: Model-based evaluation prompt for summarization recall score. here we show the demonstrations for Multi-LexSum.

```
Please act as an impartial judge and evaluate the quality of the
provided summary of a civil lawsuit.  The summary is based on a set of
legal documents, and it should contain a short description of the
background, the parties involved, and the outcomes of the case.  The
text should contain all the major points in the expert-written summary,
which are given to you.
Below is your grading rubric:  Recall:  - Evaluate the provided summary
by deciding if each of the key points is present in the provided
summary.  A key point is considered present if its factual information
is well-supported by the provided summary.  - Score:  the number of key
points present in the provided summary.  - Examples:  use the following
examples to guide your evaluation.
Example 1:
Key points:  1.  The case challenged curfews in Los Angeles and San
Bernardino, California.  2.  The curfews were issued in response to the
nationwide protests following the police killing of George Floyd in
Minneapolis.  3.  The complaint argued that the curfews violated free
speech, free assembly, free movement, and Due Process.  4.  The
complaint also argued that the San Bernardino curfew violated the
Establishment Clause.  5.  The complaint sought injunctive and
declaratory relief.  6.  The plaintiffs voluntarily dismissed the case
on July 7, 2020.  7.  The dismissal occurred because the city had
rescinded the curfews and not attempted to reinstate them.
Summary:  ''In June 2020, Black Lives Matter - Los Angeles and several
individuals filed a lawsuit in the U.S. District Court for the Central
District of California against Los Angeles Mayor Eric Garcetti, other
city officials, and the City of San Bernardino, challenging the
constitutionality of curfew orders imposed during protests against
police violence.  The plaintiffs, represented by the ACLU of Southern
California, argued that the curfews violated their First Amendment
rights to free speech and assembly, as well as their freedom of
movement, by suppressing political protests and other activities.  The
lawsuit also claimed that the curfews were not narrowly tailored to
address any emergency and lacked sufficient notice.  However, the
plaintiffs voluntarily dismissed the case in July 2020 after the
defendants lifted the curfew orders and did not reinstate them in the
following weeks.''
Reasoning:  The summary states that the plaintiffs challenged the
constitutionality of curfew orders against Los Angeles and San
Bernadino, so key point 1 is present.  The summary does not mention that
the curfew orders were issued in response to the nationwide protest that
resulted from the police killing of George Floyd in Minneapolis, so key
point 2 is missing.  The summary does mention that the complaint argued
that the curfews violated the First Amendment rights to free speech and
assembly, so key point 3 is present.  The summary does not mention that
the complaint argued that the San Bernardino curfew violated the
Establishment Clause, so key point 4 is missing.  The summary does not
mention that the complaint sought injunctive and declaratory relief, so
key point 5 is missing.  The summary mentions that the plaintiffs
voluntarily dismissed the case in July 2020 after the defendants lifted
the curfew orders and did not reinstate them in the following weeks, so
key point 6 and 7 are present.  Finally, key points 1, 3, 6, and 7 are
present in the summary, so the recall score is 4.
Output:  {{''recall'':  4}}
Example 2:  [omitted for brevity]
Now, read the provided summary and key points, and evaluate the summary
using the rubric.  First, think step-by-step and provide your reasoning
and assessment on the answer.  Then output your score in the following
json format:  {{''recall'':  2}}.
Key points:  {keypoints}
Summary:  ''{summary}''
```

Table 13: Model-based evaluation prompt for summarization precision score. Here we show the demonstrations for Multi-LexSum.

---

Please act as an impartial judge and evaluate the quality of the provided summary of a civil lawsuit. The summary is based on a set of legal documents, and it should contain a short description of the background, the parties involved, and the outcomes of the case. Below is your grading rubric: Precision: – Evaluate the provided summary by deciding if each sentence in the provided summary is supported by the information provided in the expert summary. A sentence is considered supported if its major facts align with the information in the expert summary. A sentence is still considered supported even if some of its minor details, such as dates, entity names, or the names of laws and previous court cases, are not explicitly mentioned in the expert summary. A sentence is not supported if its major facts are not mentioned or contradicted in the expert summary. – Score: the number of sentences in the provided summary that are supported by the expert summary. – Examples: use the following examples to guide your evaluation.
Example 1:
Expert summary: ''This lawsuit, brought in the the U.S. District Court for the Central District of California, was filed on June 3, 2020. The plaintiffs were represented by attorneys from the ACLU of Southern California. This lawsuit followed nation-wide protests that occurred in response to the killing of George Floyd by a police officer in Minneapolis. While most protests were peaceful, some ended in violence, property destruction, rioting, and looting. Many cities, including Los Angeles and San Bernardino, issued curfews in an attempt to quell these riots. [omitted for brevity]''
Provided summary: ''In June 2020, Black Lives Matter – Los Angeles and several individuals filed a lawsuit in the U.S. District Court for the Central District of California against Los Angeles Mayor Eric Garcetti, other city officials, and the City of San Bernardino, challenging the constitutionality of curfew orders imposed during protests against police violence. The plaintiffs, represented by the ACLU of Southern California, argued that the curfews violated their First Amendment rights to free speech and assembly, as well as their freedom of movement, by suppressing political protests and other activities. The lawsuit also claimed that the curfews were not narrowly tailored to address any emergency and lacked sufficient notice. However, the plaintiffs voluntarily dismissed the case in July 2020 after the defendants lifted the curfew orders and did not reinstate them in the following weeks.''
Reasoning: The first sentence in the provided summary is well supported by the expert summary even though some entity names are not explicitly mentioned. The second sentence is also well supported by the expert summary, as it mentions the ACLU of Southern California and the First Amendment rights. The third sentence is not supported by the expert summary, as it does not mention the lack of narrow tailoring or sufficient notice. The fourth sentence is well supported by the expert summary, as it mentions the voluntary dismissal of the case in July 2020. Therefore, the precision score is 3.
Output: {{''precision'': 3, ''sentence_count'': 4}}
Example 2: [omitted for brevity]
Now, read the provided summary and expert summary, and evaluate the summary using the rubric. First, think step-by-step and provide your reasoning and assessment on the answer. Then output your score in the following json format: {{''precision'': 2, ''sentence_count'': 6}}.
Expert summary: ''{expert_summary}''
Provided summary: ''{summary}''

---

Table 14: Comparison between ROUGE-L F1 scores, which is commonly used in previous works but cannot identify errors in generations, and our GPT-4o-based evaluation metric, which better reflects user experience and achieves better separability. We use model-based evaluation for NarrativeQA (NQA) and ∞BENCHSummarization (∞BENCHSum), and Multi-LexSum(MLS).

| Model | NarrativeQA | | ∞BENCH Sum | | Multi-LexSum | |
|---|---|---|---|---|---|---|
| | F1 | GPT | R-L | GPT | R-L | GPT |
| GPT-4o-05 | **46.5** | **55.3** | 17.4 | **34.8** | 25.5 | 55.4 |
| GPT-4o-08 | 43.1 | 51.3 | 16.8 | 31.1 | 24.9 | 56.1 |
| Claude-3.5-sonnet | 16.2 | 43.3 | 14.2 | 30.5 | 22.0 | 50.9 |
| Gemini-1.5-Flash | 39.0 | 42.9 | 17.0 | 28.7 | 24.5 | 51.1 |
| Gemini-1.5-Pro | 42.8 | 50.9 | 16.1 | 32.0 | **25.8** | **58.1** |
| Llama-3-8B-Inst | 1.0 | 6.7 | 6.8 | 0.0 | 8.0 | 3.3 |
| Llama-3-8B-Inst-Theta | 3.3 | 4.1 | 7.6 | 0.0 | 12.4 | 22.3 |
| Llama-3-70B-Inst-Theta | 4.9 | 10.9 | 5.9 | 0.0 | 14.4 | 24.0 |
| Llama-3.1-8B-Inst | 35.1 | 47.7 | 17.4 | 16.2 | 25.2 | 46.9 |
| Llama-3.1-70B-Inst | 38.3 | 54.0 | **17.5** | 19.8 | 25.2 | 51.6 |
| Llama-3.2-1B-Inst | 14.5 | 18.2 | 16.6 | 2.9 | 23.1 | 20.3 |
| Llama-3.2-3B-Inst | 23.8 | 36.7 | 16.5 | 12.4 | 23.7 | 43.6 |
| Mistral-7B-Inst-v0.1 | 12.3 | 16.0 | 12.6 | 4.2 | 22.1 | 16.1 |
| Mistral-7B-Inst-v0.2 | 14.9 | 21.0 | 13.4 | 1.0 | 20.0 | 23.4 |
| Mistral-7B-Inst-v0.3 | 21.0 | 22.0 | 12.3 | 0.1 | 19.0 | 24.4 |

Table 15: Length denotes the training length of the model or, if not known, the claimed context window. For MoE models, we denote number of active/total parameters. We change the RoPE (Su et al., 2021) $\theta$ during inference for some models and is denoted with -$\theta$. Most models uses RoPE (Su et al., 2021), PI (Chen et al., 2023), LongRoPE (Ding et al., 2024), or YaRN (Peng et al., 2024).

| Name | Length | Architecture | Positional Emb. | # Params | Inst.? |
|---|---|---|---|---|---|
| *Proprietary* | | | | | |
| gpt-4-0125-preview | 128000 | ? | ? | ? | ✓ |
| gpt-4o-2024-05-13 | 128000 | ? | ? | ? | ✓ |
| gpt-4o-2024-08-06 | 128000 | ? | ? | ? | ✓ |
| claude-3-5-sonnet-20240620 | 200000 | ? | ? | ? | ✓ |
| gemini-1.5-flash-001 | 1048576 | ? | ? | ? | ✓ |
| gemini-1.5-pro-001 | 2097152 | ? | ? | ? | ✓ |
| *Llama-2-based* (Touvron et al., 2023) | | | | | |
| LLaMA-2-7B-32K | 32768 | Full-attention | PI | 7B | ✓ |
| Llama-2-7B-32K-Instruct | 32768 | Full-attention | PI | 7B | ✓ |
| llama-2-7b-80k-basefixed | 80000 | Full-attention | Dynamic NTK | 7B | ✗ |
| Yarn-Llama-2-7b-64k | 65536 | Full-attention | YaRN | 7B | ✗ |
| Yarn-Llama-2-7b-128k | 131072 | Full-attention | YaRN | 7B | ✗ |
| *Llama-3-based* (Dubey et al., 2024) | | | | | |
| Llama-3-8B | 8192 | Full-attention | RoPE | 8B | ✗ |
| Llama-3-8B-Instruct | 8192 | Full-attention | RoPE | 8B | ✓ |
| Llama-3-8B-$\theta$ | 8192 | Full-attention | RoPE | 8B | ✗ |
| Llama-3-8B-Instruct-$\theta$ | 8192 | Full-attention | RoPE | 8B | ✓ |
| Llama-3-70B-$\theta$ | 8192 | Full-attention | RoPE | 70B | ✗ |
| Llama-3-70B-Instruct-$\theta$ | 8192 | Full-attention | RoPE | 70B | ✓ |
| Llama-3.1-8B | 131072 | Full-attention | RoPE | 8B | ✗ |
| Llama-3.1-8B-Instruct | 131072 | Full-attention | RoPE | 8B | ✓ |
| Llama-3.1-70B | 131072 | Full-attention | RoPE | 70B | ✗ |
| Llama-3.1-70B-Instruct | 131072 | Full-attention | RoPE | 70B | ✓ |
| Llama-3.3-70B-Instruct | 131072 | Full-attention | RoPE | 70B | ✓ |
| Llama-3.2-1B | 131072 | Full-attention | RoPE | 1B | ✗ |
| Llama-3.2-1B-Instruct | 131072 | Full-attention | RoPE | 1B | ✓ |
| Llama-3.2-3B | 131072 | Full-attention | RoPE | 3B | ✗ |
| Llama-3.2-3B-Instruct | 131072 | Full-attention | RoPE | 3B | ✓ |
| *ProLong* (Gao et al., 2024) | | | | | |
| ProLong | 524288 | Full-attention | RoPE | 8B | ✓ |
| *Mistral-based* (Jiang et al., 2023) | | | | | |
| Mistral-7B-Instruct-v0.1 | 8192 | Sliding window | RoPE | 7B | ✓ |
| Mistral-7B-Instruct-v0.2 | 32768 | Full-attention | RoPE | 7B | ✓ |
| Mistral-7B-v0.3 | 32768 | Full-attention | RoPE | 7B | ✗ |
| Mistral-7B-Instruct-v0.3 | 32768 | Full-attention | RoPE | 7B | ✓ |
| Mistral-Nemo-Base-2407 | 128000 | Full-attention | RoPE | 12B | ✗ |
| Mistral-Nemo-Instruct-2407 | 128000 | Full-attention | RoPE | 12B | ✓ |
| MegaBeam-Mistral-7B-512K | 524288 | Full-attention | RoPE | 7B | ✓ |
| Ministral-8B-Instruct-2410 | 131072 | Full-attn + Sliding window | RoPE | 8B | ✓ |
| *Yi* (AI et al., 2024) | | | | | |
| Yi-6B-200K | 200000 | Full-attention | RoPE | 6B | ✗ |
| Yi-9B-200K | 200000 | Full-attention | RoPE | 9B | ✗ |
| Yi-34B-200K | 200000 | Full-attention | RoPE | 34B | ✗ |
| Yi-1.5-9B-32K | 32768 | Full-attention | RoPE | 9B | ✗ |
| *Phi-3* (Abdin et al., 2024) | | | | | |
| Phi-3-mini-128k-instruct | 131072 | Full-attention | LongRoPE | 4B | ✓ |
| Phi-3-small-128k-instruct | 131072 | Blocksparse attention | LongRoPE | 7B | ✓ |
| Phi-3-medium-128k-instruct | 131072 | Full-attention | LongRoPE | 14B | ✓ |
| Phi-3.5-mini-instruct | 131072 | Full-attention | LongRoPE | 4B | ✓ |
| *Qwen2* (Yang et al., 2024) | | | | | |
| Qwen2-7B | 32768 | Full-attention | YaRN | 7B | ✗ |
| Qwen2-7B-Instruct | 32768 | Full-attention | YaRN | 7B | ✓ |
| Qwen2-57B-A14B | 32768 | Full-attention MoE | YaRN | 14B/57B | ✗ |
| Qwen2-57B-A14B-Instruct | 32768 | Full-attention MoE | YaRN | 14B/57B | ✓ |
| *Qwen2.5* (Qwen et al., 2025) | | | | | |
| Qwen2.5-1.5B | 131072 | Dual-Chunk Attention | YaRN | 1.5B | ✗ |
| Qwen2.5-1.5B-Inst | 131072 | Dual-Chunk Attention | YaRN | 1.5B | ✓ |
| Qwen2.5-3B | 131072 | Dual-Chunk Attention | YaRN | 3B | ✗ |
| Qwen2.5-3B-Inst | 131072 | Dual-Chunk Attention | YaRN | 3B | ✓ |
| Qwen2.5-7B | 131072 | Dual-Chunk Attention | YaRN | 7B | ✗ |
| Qwen2.5-7B-Inst | 131072 | Dual-Chunk Attention | YaRN | 7B | ✓ |
| Qwen2.5-72B-Inst | 131072 | Dual-Chunk Attention | YaRN | 72B | ✓ |
| *Qwen2.5-1M* (Yang et al., 2025) | | | | | |
| Qwen2.5-7B-Inst-1M | 1,010,000 | Dual-Chunk Attention | YaRN | 7B | ✓ |
| Qwen2.5-14B-Inst-1M | 1,010,000 | Dual-Chunk Attention | YaRN | 14B | ✓ |
| *Jamba* (Lieber et al., 2024) (Team et al., 2024c) | | | | | |
| Jamba-v0.1 | 262144 | Hybrid (Mamba + Full-attention) MoE | None | 12B/52B | ✗ |
| AI21-Jamba-1.5-Mini | 262144 | Hybrid (Mamba + Full-attention) MoE | None | 12B/52B | ✓ |

| Model | Recall 8K | 16K | 32K | 64K | 128K | RAG 8K | 16K | 32K | 64K | 128K | Cite 8K | 16K | 32K | 64K | 128K | Re-rank 8K | 16K | 32K | 64K | 128K |
|---|---|---|---|---|---|---|---|---|---|---|---|---|---|---|---|---|---|---|---|---|
| GPT-4 | 99.5 | 99.8 | 97.6 | 91.2 | 73.5 | 75.1 | 73.5 | 70.9 | 68.1 | 64.7 | 44.0 | 42.7 | 26.1 | 3.1 | 1.2 | 87.8 | 78.1 | 54.7 | 25.8 | 9.6 |
| GPT-4o-05 | 93.2 | 92.2 | 91.2 | 90.1 | 82.4 | 72.5 | 72.4 | 72.6 | 72.7 | 70.4 | 44.2 | 43.7 | 42.3 | 42.4 | 40.4 | 86.3 | 77.9 | 67.7 | 57.7 | 47.7 |
| GPT-4o-08 | 100.0 | 100.0 | 100.0 | 100.0 | 99.9 | 73.9 | 74.0 | 72.4 | 71.5 | 70.2 | 45.4 | 46.7 | 45.4 | 45.4 | 44.3 | 86.9 | 80.5 | 68.5 | 58.4 | 50.0 |
| GPT-4o-mini | 99.8 | 99.8 | 99.1 | 96.1 | 89.6 | 72.6 | 72.1 | 70.0 | 68.7 | 68.0 | 36.1 | 33.7 | 30.0 | 27.1 | 24.3 | 80.8 | 71.1 | 58.8 | 42.0 | 31.2 |
| Claude-3.5-Sonnet | 99.5 | 96.6 | 96.2 | 97.2 | 94.7 | 55.4 | 35.8 | 33.4 | 32.5 | 38.1 | 34.8 | 32.2 | 28.3 | 26.3 | 18.7 | 75.2 | 41.0 | 34.8 | 17.9 | 7.2 |
| Gemini-1.5-Flash | 93.5 | 93.8 | 93.7 | 93.2 | 91.2 | 71.9 | 70.5 | 69.2 | 68.0 | 66.5 | 48.7 | 49.4 | 48.1 | 40.2 | 31.7 | 87.8 | 80.8 | 74.0 | 61.2 | 50.8 |
| Gemini-1.5-Pro | 87.5 | 87.3 | 96.3 | 87.2 | 91.0 | 73.6 | 73.4 | 72.3 | 71.8 | 71.1 | 47.2 | 45.9 | 44.8 | 46.2 | 43.6 | 87.6 | 84.1 | 76.5 | 68.4 | 59.7 |
| Llama-3-8B-θ | 93.2 | 77.6 | 53.2 | 16.9 | 0.0 | 59.8 | 60.4 | 58.8 | 39.2 | 2.4 | 15.1 | 11.5 | 5.4 | 3.5 | 0.1 | 44.4 | 34.4 | 30.6 | 14.8 | 0.0 |
| Llama-3-8B-θ | 97.3 | 91.8 | 67.6 | 39.4 | 0.0 | 65.9 | 64.2 | 61.5 | 48.1 | 1.3 | 22.4 | 20.7 | 13.1 | 4.8 | 0.8 | 59.5 | 37.1 | 26.5 | 14.4 | 0.1 |
| Llama-3-70B-θ | 91.0 | 85.2 | 65.2 | 36.9 | 0.0 | 68.1 | 65.2 | 62.5 | 55.5 | 0.0 | 23.8 | 22.2 | 16.3 | 6.7 | 2.9 | 54.7 | 44.1 | 41.2 | 20.3 | 0.0 |
| Llama-3-70B-θ | 97.5 | 95.0 | 81.2 | 48.6 | 0.0 | 70.5 | 68.3 | 66.6 | 60.4 | 0.0 | 43.5 | 44.2 | 37.7 | 7.6 | 0.0 | 82.1 | 73.8 | 59.0 | 36.2 | 0.0 |
| Llama-3.1-8B | 97.3 | 92.9 | 90.1 | 86.6 | 76.6 | 65.8 | 65.5 | 63.8 | 61.5 | 54.5 | 17.3 | 10.3 | 5.1 | 2.6 | 1.6 | 46.6 | 38.3 | 34.9 | 22.7 | 9.3 |
| Llama-3.1-8B | 100.0 | 100.0 | 99.7 | 99.2 | 95.2 | 70.8 | 68.8 | 66.5 | 65.8 | 59.5 | 36.3 | 26.7 | 10.9 | 13.1 | 2.9 | 67.0 | 51.7 | 43.4 | 32.8 | 14.0 |
| Llama-3.1-70B | 99.8 | 99.8 | 99.1 | 92.9 | 76.3 | 69.2 | 68.2 | 67.5 | 65.2 | 56.6 | 23.2 | 21.7 | 16.2 | 9.7 | 3.9 | 57.7 | 53.0 | 43.7 | 36.9 | 22.6 |
| Llama-3.1-70B | 100.0 | 100.0 | 99.9 | 98.5 | 90.7 | 74.0 | 71.5 | 70.9 | 69.5 | 56.2 | 44.3 | 41.3 | 40.1 | 29.8 | 7.5 | 83.5 | 73.5 | 60.2 | 41.0 | 24.5 |
| Llama-3.2-1B | 73.5 | 60.6 | 41.8 | 43.4 | 25.2 | 41.5 | 39.1 | 37.6 | 35.5 | 32.1 | 6.6 | 5.6 | 5.4 | 2.9 | 1.5 | 32.4 | 26.2 | 18.9 | 9.4 | 5.6 |
| Llama-3.2-1B | 72.9 | 62.6 | 46.9 | 49.2 | 32.4 | 48.7 | 44.6 | 43.7 | 37.9 | 33.8 | 6.3 | 5.8 | 4.5 | 2.8 | 2.4 | 26.0 | 18.5 | 5.7 | 3.9 | 1.7 |
| Llama-3.2-3B | 90.8 | 89.3 | 79.6 | 65.8 | 58.8 | 61.1 | 60.9 | 58.8 | 55.3 | 51.4 | 10.4 | 7.1 | 3.5 | 3.2 | 2.1 | 43.6 | 33.5 | 26.3 | 6.9 | 5.4 |
| Llama-3.2-3B | 99.2 | 93.1 | 84.4 | 64.1 | 53.1 | 65.3 | 64.9 | 63.1 | 60.1 | 56.4 | 11.2 | 7.5 | 5.0 | 7.6 | 7.4 | 41.0 | 29.4 | 17.7 | 3.9 | 0.7 |
| Llama-7B-80k | 85.0 | 71.1 | 54.1 | 33.8 | 17.0 | 55.2 | 53.0 | 52.1 | 50.7 | 40.8 | 12.4 | 8.4 | 6.3 | 4.2 | 1.0 | 37.0 | 26.5 | 19.6 | 9.1 | 0.2 |
| Yarn-Llama-2-7B-128k | 58.4 | 39.2 | 17.5 | 12.3 | 4.2 | 53.6 | 50.7 | 51.3 | 42.3 | 26.7 | 5.3 | 4.5 | 3.5 | 2.5 | 0.7 | 43.2 | 18.3 | 11.2 | 4.4 | 0.4 |
| Mistral-7B-v0.1 | 35.6 | 16.9 | 10.4 | 6.6 | 3.8 | 52.2 | 46.8 | 43.0 | 38.6 | 36.6 | 4.7 | 2.3 | 2.8 | 1.0 | 1.3 | 2.8 | 0.6 | 0.2 | 0.0 | 0.0 |
| Mistral-7B-v0.2 | 93.8 | 90.6 | 74.9 | 28.5 | 1.5 | 63.7 | 60.6 | 58.4 | 40.6 | 11.4 | 22.7 | 8.6 | 6.2 | 2.3 | 0.6 | 62.1 | 44.0 | 29.4 | 1.2 | 1.2 |
| Mistral-7B-v0.3 | 96.9 | 95.2 | 86.1 | 25.6 | 3.6 | 62.8 | 61.5 | 57.2 | 42.9 | 4.2 | 14.1 | 10.8 | 5.2 | 1.3 | 0.4 | 42.4 | 34.8 | 22.6 | 0.4 | 1.2 |
| Mistral-7B-v0.3 | 96.6 | 94.8 | 80.8 | 34.7 | 8.0 | 64.9 | 63.5 | 61.2 | 47.4 | 21.3 | 32.9 | 16.0 | 8.0 | 2.4 | 0.9 | 64.0 | 49.3 | 40.0 | 2.5 | 0.5 |
| Mistral-Nemo | 99.9 | 95.6 | 65.6 | 30.1 | 14.6 | 67.8 | 63.5 | 57.3 | 47.1 | 40.0 | 32.7 | 8.2 | 2.8 | 1.1 | 0.5 | 66.3 | 48.3 | 14.8 | 0.0 | 0.0 |
| MegaBeam-Mistral | 97.8 | 97.1 | 95.9 | 94.2 | 89.6 | 64.7 | 63.0 | 61.8 | 59.7 | 57.0 | 22.0 | 14.1 | 9.0 | 4.5 | 4.0 | 56.4 | 40.3 | 35.0 | 25.6 | 14.7 |
| Ministral-8B | 98.9 | 97.7 | 95.1 | 30.9 | 12.8 | 66.7 | 66.0 | 63.5 | 47.1 | 35.2 | 36.7 | 24.1 | 7.6 | 0.7 | 0.4 | 67.4 | 49.4 | 32.6 | 0.0 | 0.0 |
| Phi-3-mini-128k | 97.8 | 96.8 | 95.1 | 95.4 | 50.1 | 62.5 | 61.4 | 59.4 | 56.3 | 46.7 | 23.3 | 18.5 | 10.1 | 2.1 | 0.6 | 53.1 | 33.1 | 23.5 | 14.0 | 5.8 |
| Phi-3-small-128k | 94.0 | 93.6 | 83.4 | 74.6 | 22.3 | 67.1 | 64.7 | 62.6 | 58.5 | 33.8 | 17.9 | 13.5 | 6.9 | 3.5 | 3.0 | 46.1 | 35.5 | 27.4 | 16.9 | 1.9 |
| Phi-3-med-128k | 91.6 | 88.9 | 79.3 | 69.3 | 24.5 | 66.1 | 63.8 | 62.3 | 57.8 | 44.8 | 39.8 | 26.4 | 9.7 | 5.3 | 3.3 | 52.6 | 36.7 | 26.1 | 12.9 | 6.6 |
| Phi-3.5-mini | 98.3 | 97.8 | 94.7 | 91.9 | 48.8 | 61.9 | 60.6 | 57.1 | 52.5 | 43.1 | 22.2 | 15.8 | 7.2 | 2.0 | 1.6 | 51.3 | 36.4 | 25.6 | 14.2 | 7.8 |
| Yi-6B-200k | 85.4 | 72.4 | 54.7 | 38.3 | 37.4 | 49.2 | 45.6 | 42.2 | 39.7 | 36.9 | 4.1 | 2.3 | 0.9 | 0.8 | 1.1 | 43.9 | 26.7 | 11.8 | 5.5 | 3.4 |
| Yi-9B-200k | 95.5 | 90.4 | 77.9 | 70.5 | 56.2 | 61.4 | 60.8 | 60.7 | 55.5 | 50.5 | 12.5 | 10.0 | 5.8 | 5.0 | 3.3 | 44.7 | 35.3 | 26.1 | 15.6 | 8.3 |
| Yi-34B-200k | 95.9 | 91.2 | 89.9 | 82.9 | 74.1 | 68.2 | 65.9 | 64.5 | 63.0 | 59.5 | 9.5 | 7.4 | 3.4 | 1.5 | 1.7 | 55.1 | 45.6 | 34.6 | 27.2 | 20.5 |
| Qwen2-7B | 73.2 | 65.4 | 66.1 | 46.2 | 36.8 | 58.5 | 57.3 | 54.0 | 51.1 | 47.4 | 15.5 | 7.4 | 3.6 | 3.5 | 2.4 | 47.9 | 28.2 | 19.5 | 8.6 | 5.9 |
| Qwen2-57B | 98.6 | 96.3 | 90.2 | 26.6 | 5.9 | 65.4 | 66.2 | 62.2 | 54.7 | 12.7 | 36.0 | 11.3 | 4.1 | 3.9 | 1.1 | 53.3 | 38.9 | 22.3 | 6.5 | 0.0 |
| Qwen2.5-7B | 98.9 | 97.9 | 95.2 | 47.8 | 11.4 | 64.2 | 64.8 | 57.7 | 46.4 | 30.6 | 36.6 | 27.2 | 16.5 | 4.9 | 3.1 | 57.9 | 46.8 | 34.7 | 11.3 | 1.6 |
| Qwen2.5-72B | 100.0 | 100.0 | 98.4 | 71.4 | 38.4 | 72.5 | 72.0 | 70.2 | 63.1 | 43.0 | 42.7 | 40.8 | 39.0 | 25.1 | 8.0 | 86.0 | 75.3 | 60.3 | 41.0 | 24.5 |
| Qwen2.5-7B | 98.9 | 97.9 | 95.2 | 47.8 | 11.4 | 64.2 | 64.8 | 57.7 | 46.4 | 30.6 | 36.6 | 27.2 | 16.5 | 4.9 | 3.1 | 57.9 | 46.8 | 34.7 | 11.3 | 1.6 |
| Qwen2.5-72B | 100.0 | 100.0 | 98.4 | 71.4 | 38.4 | 72.5 | 72.0 | 70.2 | 63.1 | 43.0 | 42.7 | 40.8 | 39.0 | 25.1 | 8.0 | 86.0 | 75.3 | 60.3 | 41.0 | 24.5 |
| Qwen2.5-7B-1M | 96.9 | 98.5 | 94.1 | 92.8 | 86.0 | 67.7 | 68.3 | 64.9 | 64.0 | 59.0 | 36.2 | 17.9 | 8.6 | 9.6 | 4.8 | 71.3 | 57.4 | 45.9 | 32.1 | 24.8 |
| Qwen2.5-14B-1M | 100.0 | 99.9 | 99.4 | 99.2 | 97.8 | 72.2 | 70.8 | 68.9 | 66.6 | 62.0 | 41.9 | 35.4 | 32.9 | 28.2 | 27.1 | 74.3 | 57.9 | 49.9 | 38.0 | 27.9 |
| Jamba-1.5-Mini | 98.4 | 95.4 | 97.3 | 93.2 | 90.0 | 65.2 | 64.1 | 63.4 | 61.8 | 57.3 | 14.4 | 8.3 | 6.6 | 6.4 | 3.1 | 62.1 | 46.3 | 33.4 | 25.4 | 14.6 |
| ProLong | 99.4 | 99.7 | 99.2 | 99.4 | 98.8 | 67.4 | 67.5 | 66.4 | 65.7 | 63.2 | 34.6 | 27.1 | 12.8 | 5.8 | 1.4 | 60.4 | 47.4 | 40.2 | 31.1 | 22.5 |

| Model | LongQA 8K | 16K | 32K | 64K | 128K | Summ 8K | 16K | 32K | 64K | 128K | ICL 8K | 16K | 32K | 64K | 128K | Avg. 8K | 16K | 32K | 64K | 128K |
|---|---|---|---|---|---|---|---|---|---|---|---|---|---|---|---|---|---|---|---|---|
| GPT-4 | 46.6 | 46.1 | 46.7 | 44.2 | 46.7 | 28.2 | 29.1 | 28.6 | 32.8 | 33.0 | 80.8 | 82.0 | 76.5 | 59.0 | 46.0 | 66.0 | 64.5 | 57.3 | 46.3 | 39.2 |
| GPT-4o-05 | 43.8 | 48.0 | 53.6 | 54.3 | 62.0 | 29.5 | 33.2 | 39.4 | 40.8 | 43.6 | 79.9 | 80.1 | 75.8 | 63.0 | 48.4 | 64.2 | 63.9 | 63.3 | 60.1 | 56.4 |
| GPT-4o-08 | 41.8 | 48.1 | 50.8 | 55.5 | 59.3 | 28.3 | 34.3 | 38.9 | 40.1 | 43.2 | 82.6 | 85.9 | 87.2 | 86.8 | 86.3 | 65.5 | 67.1 | 66.2 | 65.4 | 64.8 |
| GPT-4o-mini | 35.3 | 39.8 | 45.8 | 49.3 | 54.8 | 26.7 | 33.8 | 35.1 | 39.0 | 40.8 | 77.3 | 79.9 | 81.9 | 81.9 | 81.2 | 61.2 | 61.4 | 60.1 | 57.7 | 55.7 |
| Claude-3.5-Sonnet | 29.2 | 23.1 | 27.2 | 12.6 | 12.6 | 28.9 | 33.8 | 36.5 | 35.6 | 36.6 | 87.1 | 88.7 | 89.9 | 89.2 | 61.0 | 58.6 | 50.2 | 49.5 | 44.5 | 38.4 |
| Gemini-1.5-Flash | 33.9 | 42.4 | 46.3 | 52.3 | 57.3 | 25.0 | 28.4 | 33.5 | 38.3 | 39.4 | 74.2 | 68.7 | 56.6 | 39.7 | 28.1 | 62.1 | 62.0 | 60.2 | 56.1 | 52.1 |
| Gemini-1.5-Pro | 35.5 | 41.5 | 49.3 | 52.6 | 59.6 | 30.0 | 32.7 | 36.3 | 41.8 | 46.4 | 78.7 | 81.1 | 82.0 | 80.4 | 79.4 | 62.9 | 63.7 | 65.4 | 64.0 | 64.4 |
| Llama-3-8B-θ | 19.8 | 24.7 | 25.8 | 1.5 | 0.8 | 0.8 | 0.5 | 1.1 | 0.5 | 0.0 | 53.9 | 64.0 | 68.0 | 73.9 | 2.8 | 41.0 | 39.0 | 34.7 | 21.5 | 0.9 |
| Llama-3-8B-θ | 24.6 | 26.2 | 30.2 | 10.9 | 2.5 | 23.1 | 26.7 | 29.9 | 19.4 | 1.9 | 64.8 | 71.1 | 76.0 | 79.3 | 4.3 | 51.1 | 48.2 | 43.5 | 30.9 | 1.6 |
| Llama-3-70B-θ | 31.4 | 37.3 | 41.3 | 29.6 | 5.4 | 8.0 | 9.0 | 5.6 | 2.8 | 0.1 | 67.7 | 71.4 | 70.8 | 76.6 | 0.2 | 49.2 | 47.8 | 43.3 | 32.6 | 1.2 |
| Llama-3-70B-θ | 25.4 | 33.8 | 41.1 | 26.6 | 12.0 | 26.9 | 31.6 | 37.3 | 34.6 | 2.7 | 64.5 | 68.0 | 70.4 | 75.4 | 0.0 | 58.6 | 59.3 | 56.2 | 41.3 | 2.1 |
| Llama-3.1-8B | 23.6 | 27.2 | 30.8 | 35.7 | 36.2 | 0.1 | 0.0 | 0.0 | 0.0 | 1.7 | 64.1 | 69.6 | 71.7 | 75.7 | 79.5 | 45.0 | 43.4 | 42.3 | 40.7 | 37.1 |
| Llama-3.1-8B | 25.2 | 32.6 | 40.5 | 44.6 | 43.2 | 21.3 | 24.2 | 27.5 | 26.7 | 27.0 | 71.3 | 76.4 | 77.9 | 81.4 | 83.9 | 56.0 | 54.3 | 52.3 | 52.0 | 46.5 |
| Llama-3.1-70B | 35.2 | 41.1 | 42.4 | 45.0 | 46.0 | 9.4 | 11.5 | 7.3 | 4.1 | 6.3 | 69.0 | 74.8 | 73.6 | 80.7 | 80.4 | 51.9 | 52.9 | 50.0 | 47.8 | 41.7 |
| Llama-3.1-70B | 31.2 | 40.0 | 47.0 | 54.0 | 56.3 | 26.1 | 31.3 | 34.2 | 32.9 | 31.6 | 71.6 | 75.0 | 76.1 | 81.6 | 81.4 | 61.5 | 61.8 | 61.2 | 58.2 | 49.7 |
| Llama-3.2-1B | 14.3 | 14.1 | 14.4 | 14.4 | 15.9 | 0.4 | 0.2 | 0.0 | 0.3 | 0.9 | 54.8 | 63.8 | 70.0 | 75.4 | 76.9 | 31.9 | 29.9 | 26.9 | 25.9 | 22.6 |
| Llama-3.2-1B | 16.2 | 17.6 | 19.9 | 18.5 | 18.1 | 8.0 | 9.3 | 11.6 | 10.6 | 9.1 | 56.5 | 65.4 | 70.9 | 75.0 | 77.3 | 33.5 | 32.0 | 29.0 | 28.3 | 25.0 |
| Llama-3.2-3B | 20.6 | 23.9 | 25.8 | 28.1 | 27.6 | 0.1 | 0.3 | 0.0 | 0.0 | 0.0 | 66.6 | 77.2 | 82.0 | 86.7 | 89.5 | 41.9 | 44.8 | 39.4 | 35.1 | 33.5 |
| Llama-3.2-3B | 21.5 | 26.8 | 30.4 | 32.0 | 28.9 | 18.6 | 21.2 | 23.5 | 25.3 | 25.9 | 68.6 | 76.4 | 80.4 | 84.0 | 86.4 | 46.5 | 45.6 | 43.5 | 39.6 | 37.0 |
| Llama-7B-80k | 19.4 | 24.5 | 21.8 | 24.2 | 19.8 | 0.8 | 0.7 | 0.4 | 0.2 | 0.9 | 57.2 | 64.7 | 67.5 | 70.8 | 76.8 | 38.1 | 35.6 | 31.7 | 27.6 | 22.4 |
| Yarn-Llama-2-7B-128k | 18.2 | 21.7 | 20.2 | 21.9 | 16.1 | 1.1 | 0.6 | 0.7 | 1.1 | 1.9 | 63.0 | 71.1 | 76.3 | 78.2 | 75.4 | 34.7 | 29.4 | 25.8 | 23.2 | 17.9 |
| Mistral-7B-v0.1 | 16.3 | 13.2 | 15.6 | 14.7 | 15.2 | 13.5 | 12.2 | 10.1 | 11.0 | 9.2 | 45.2 | 44.6 | 44.7 | 44.6 | 44.8 | 24.3 | 19.5 | 18.1 | 16.6 | 15.8 |
| Mistral-7B-v0.2 | 18.4 | 25.2 | 26.3 | 15.4 | 4.1 | 21.6 | 24.0 | 24.3 | 16.0 | 4.3 | 65.4 | 72.0 | 78.4 | 64.3 | 38.0 | 49.7 | 46.4 | 42.6 | 24.0 | 8.7 |
| Mistral-7B-v0.3 | 24.7 | 29.3 | 28.6 | 21.5 | 10.7 | 1.9 | 0.6 | 0.5 | 1.7 | 0.4 | 64.0 | 73.3 | 79.1 | 80.9 | 54.8 | 43.8 | 43.6 | 39.9 | 24.9 | 10.8 |
| Mistral-7B-v0.3 | 26.2 | 34.0 | 30.9 | 22.5 | 12.0 | 21.3 | 21.8 | 21.7 | 10.8 | 5.1 | 73.1 | 79.0 | 81.6 | 81.2 | 73.6 | 54.1 | 51.2 | 46.3 | 28.8 | 17.3 |
| Mistral-Nemo | 31.0 | 33.4 | 27.3 | 21.4 | 22.5 | 25.3 | 19.8 | 20.8 | 19.3 | 18.5 | 68.6 | 76.0 | 80.2 | 82.2 | 84.0 | 55.9 | 49.2 | 38.4 | 28.7 | 25.7 |
| MegaBeam-Mistral | 20.7 | 30.7 | 32.2 | 34.9 | 37.3 | 20.3 | 22.6 | 24.1 | 29.0 | 28.9 | 72.1 | 76.4 | 80.6 | 83.8 | 86.2 | 50.6 | 49.2 | 48.4 | 47.4 | 45.4 |
| Ministral-8B | 28.8 | 33.4 | 34.9 | 23.7 | 25.0 | 22.7 | 25.6 | 23.1 | 14.4 | 11.5 | 71.3 | 74.8 | 77.3 | 82.8 | 84.4 | 56.1 | 53.0 | 47.7 | 28.5 | 24.2 |
| Phi-3-mini-128k | 28.2 | 31.4 | 32.9 | 33.9 | 29.9 | 19.1 | 21.7 | 24.0 | 25.0 | 23.7 | 65.6 | 69.6 | 73.7 | 76.2 | 78.7 | 49.9 | 47.5 | 45.5 | 43.3 | 33.6 |
| Phi-3-small-128k | 25.0 | 28.5 | 36.0 | 41.8 | 27.5 | 18.6 | 20.3 | 24.8 | 24.4 | 6.6 | 67.8 | 74.1 | 79.1 | 82.3 | 79.6 | 48.1 | 47.2 | 45.7 | 43.2 | 24.9 |
| Phi-3-med-128k | 22.7 | 21.2 | 19.1 | 18.4 | 26.9 | 21.4 | 23.3 | 23.0 | 27.6 | 24.8 | 62.6 | 65.5 | 67.8 | 73.0 | 73.2 | 51.0 | 46.5 | 41.0 | 37.7 | 29.1 |
| Phi-3.5-mini | 24.7 | 26.4 | 29.3 | 30.6 | 28.5 | 19.6 | 21.1 | 22.9 | 24.9 | 26.3 | 63.8 | 70.2 | 75.2 | 78.0 | 79.5 | 48.8 | 46.9 | 44.6 | 42.0 | 33.6 |
| Yi-6B-200k | 23.9 | 24.9 | 25.7 | 28.7 | 30.0 | 1.3 | 0.9 | 3.3 | 3.4 | 3.5 | 67.0 | 71.9 | 76.0 | 78.9 | 82.3 | 39.3 | 34.9 | 30.7 | 27.9 | 27.8 |
| Yi-9B-200k | 24.2 | 29.3 | 33.9 | 36.7 | 39.3 | 2.6 | 3.8 | 3.8 | 3.2 | 6.2 | 54.3 | 58.8 | 61.4 | 68.5 | 73.4 | 42.2 | 41.2 | 38.5 | 36.4 | 33.9 |
| Yi-34B-200k | 24.3 | 24.0 | 25.4 | 26.2 | 26.6 | 11.0 | 12.1 | 11.2 | 12.1 | 14.3 | 71.8 | 76.7 | 80.6 | 82.8 | 85.0 | 48.0 | 46.1 | 44.2 | 42.2 | 40.3 |
| Qwen2-7B | 29.1 | 34.2 | 32.5 | 35.3 | 31.9 | 17.7 | 18.3 | 20.2 | 23.9 | 27.6 | 63.0 | 67.8 | 68.6 | 66.0 | 66.2 | 43.6 | 39.8 | 37.8 | 33.5 | 31.2 |
| Qwen2-57B | 32.7 | 32.1 | 34.2 | 35.2 | 22.9 | 21.3 | 23.0 | 21.9 | 25.0 | 10.5 | 73.9 | 80.9 | 83.5 | 84.4 | 40.6 | 54.5 | 49.8 | 45.5 | 33.8 | 13.4 |
| Qwen2.5-7B | 26.0 | 34.9 | 28.9 | 21.8 | 21.9 | 23.5 | 28.2 | 27.1 | 25.2 | 18.8 | 72.1 | 77.9 | 79.2 | 79.4 | 72.0 | 54.2 | 54.0 | 48.5 | 34.2 | 22.8 |
| Qwen2.5-72B | 36.2 | 40.5 | 44.4 | 41.5 | 38.9 | 28.2 | 33.9 | 38.8 | 34.1 | 31.4 | 78.4 | 80.4 | 81.8 | 83.6 | 83.2 | 63.4 | 63.3 | 61.8 | 51.4 | 38.2 |
| Qwen2.5-7B | 26.0 | 34.9 | 28.9 | 21.8 | 21.9 | 23.5 | 28.2 | 27.1 | 25.2 | 18.8 | 72.1 | 77.9 | 79.2 | 79.4 | 72.0 | 54.2 | 54.0 | 48.5 | 34.2 | 22.8 |
| Qwen2.5-72B | 36.2 | 40.5 | 44.4 | 41.5 | 38.9 | 28.2 | 33.9 | 38.8 | 34.1 | 31.4 | 78.4 | 80.4 | 81.8 | 83.6 | 83.2 | 63.4 | 63.3 | 61.8 | 51.4 | 38.2 |
| Qwen2.5-7B-1M | 28.8 | 36.7 | 37.9 | 36.5 | 38.6 | 22.9 | 25.0 | 27.5 | 27.9 | 28.9 | 70.8 | 69.2 | 57.2 | 49.4 | 47.1 | 54.6 | 53.3 | 48.0 | 44.6 | 41.3 |
| Qwen2.5-14B-1M | 31.0 | 38.1 | 39.7 | 44.1 | 47.8 | 27.3 | 32.1 | 34.3 | 35.3 | 38.6 | 79.8 | 84.7 | 88.2 | 89.6 | 90.5 | 60.9 | 59.9 | 59.0 | 57.2 | 56.0 |
| Jamba-1.5-Mini | 35.9 | 40.0 | 48.1 | 51.0 | 54.2 | 17.1 | 17.3 | 17.3 | 17.7 | 18.1 | 81.0 | 85.0 | 87.9 | 89.7 | 91.0 | 53.4 | 50.9 | 50.6 | 49.3 | 46.9 |
| ProLong | 28.3 | 36.9 | 37.6 | 40.8 | 43.9 | 21.7 | 22.3 | 22.2 | 26.0 | 29.2 | 69.1 | 75.1 | 78.9 | 83.2 | 86.5 | 54.4 | 53.7 | 51.1 | 50.3 | 49.4 |

Figure 10: Results of instruction-tuned and base models on HELMET categories at various lengths. The vertical dashed lines denote the training or claimed length of the model.

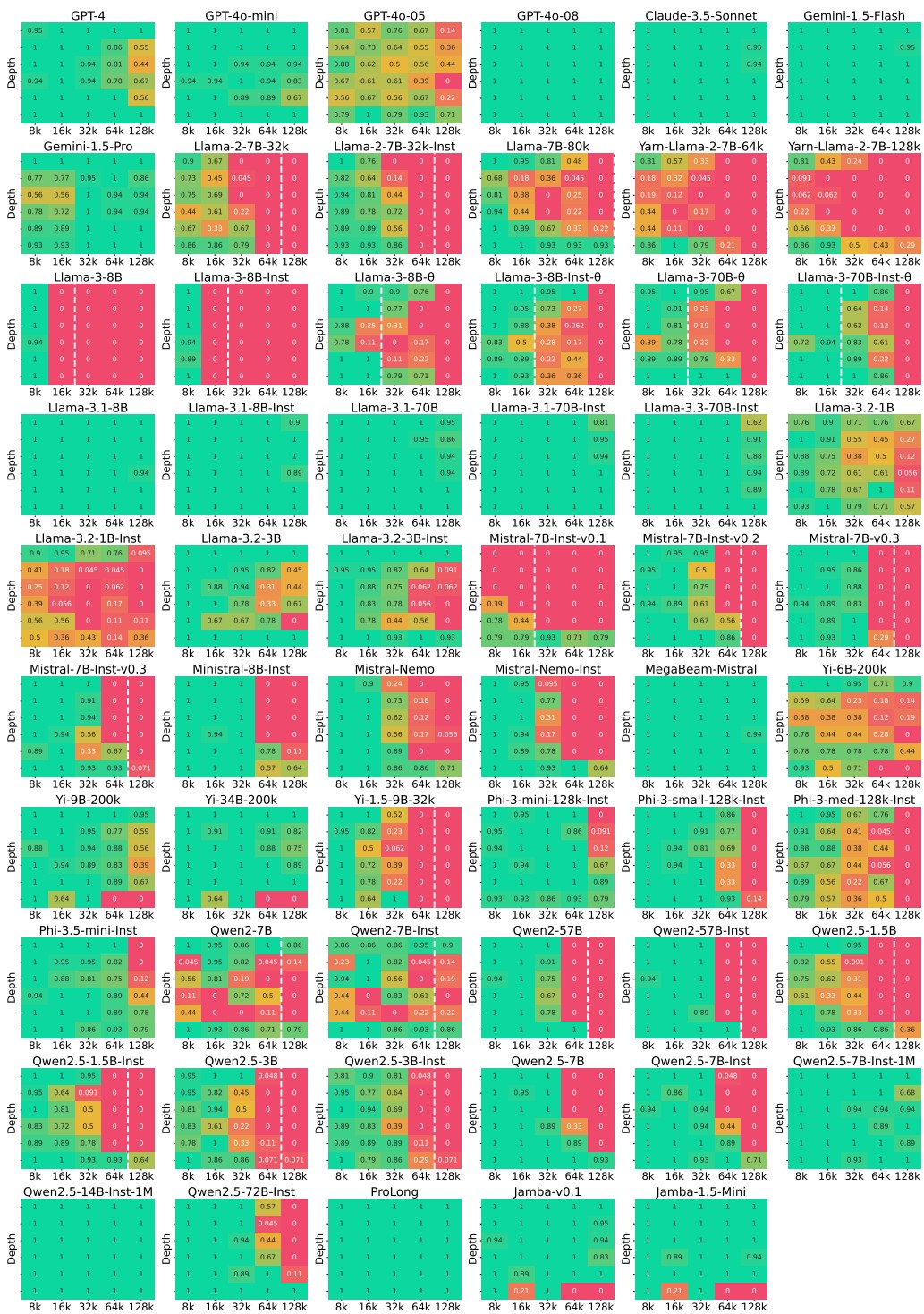

Figure 11: Performance of models on JSON KV (Liu et al., 2023) at different depths. Depth is the position of the gold KV pair, and its values are $[0.0, 0.2, 0.4, 0.6, 0.8, 1.0]$, where $0.0$ is the beginning of the context (the top of each heatmap) and $1.0$ is the end (the bottom of each heatmap).

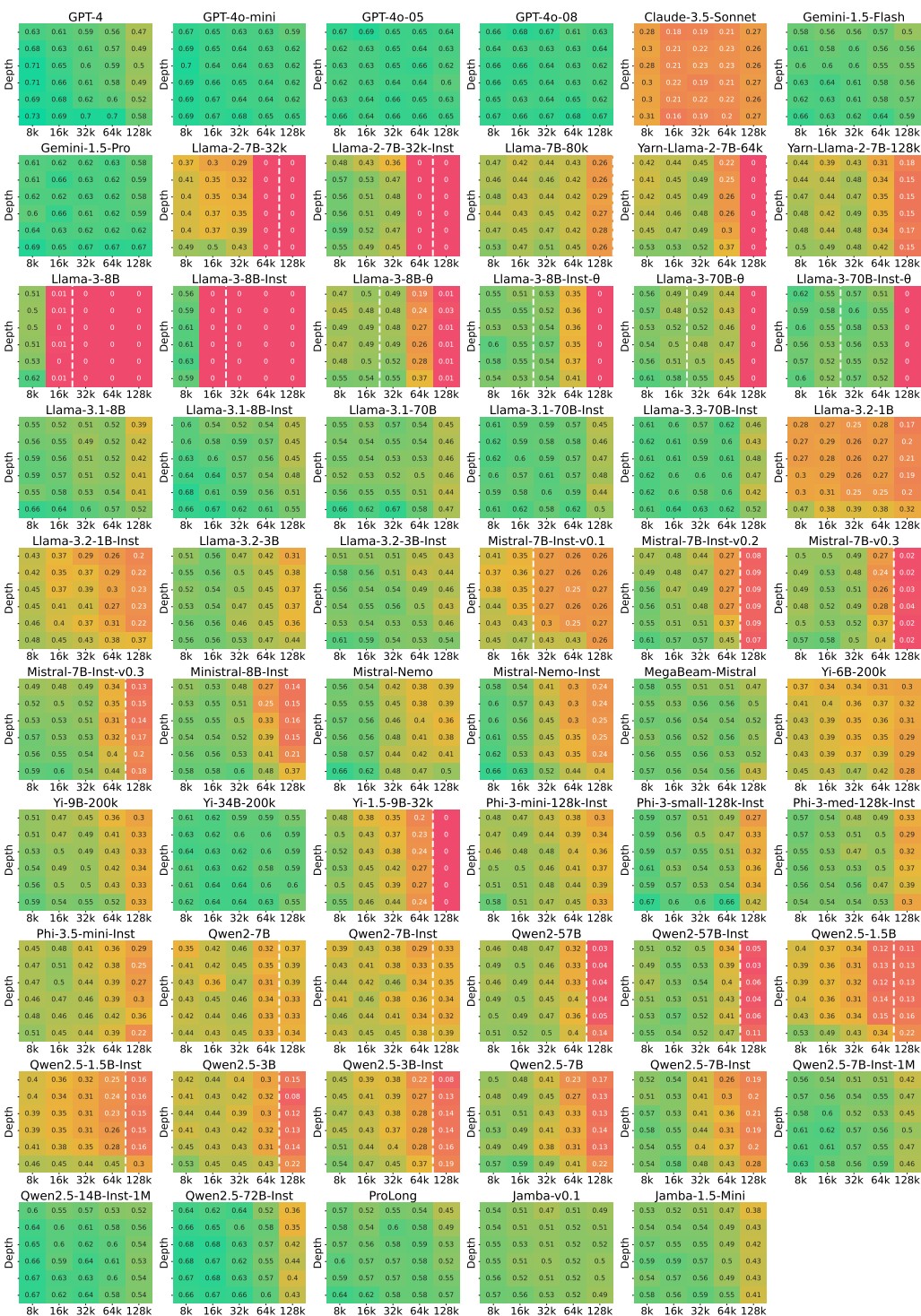

Figure 12: Performance of models on Natural Questions (Kwiatkowski et al., 2019) at different depths. Depth is the position of the gold passage, and its values are $[0.0, 0.2, 0.4, 0.6, 0.8, 1.0]$, where $0.0$ is the beginning of the context (the top of each heatmap) and $1.0$ is the end (the bottom of each heatmap).

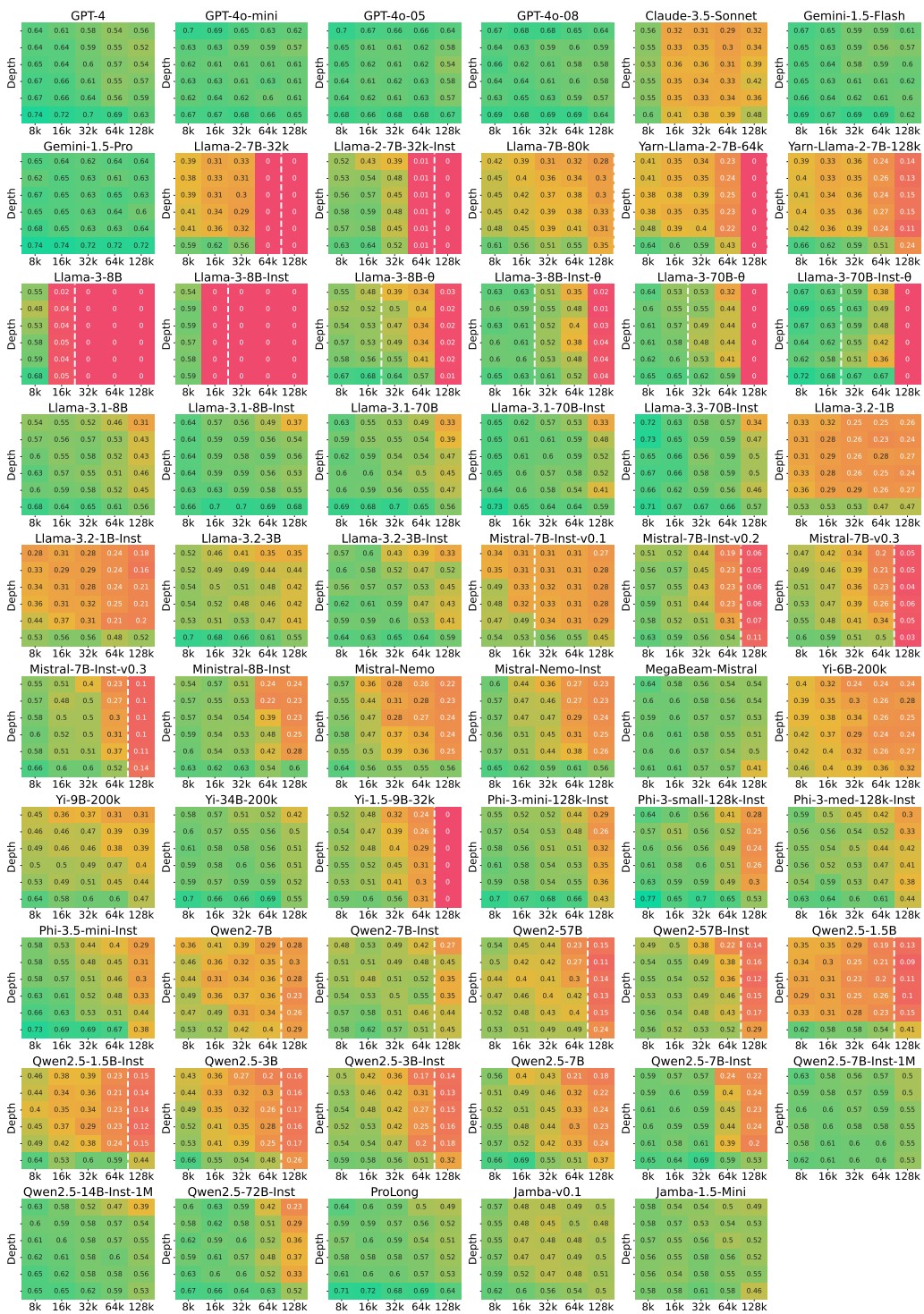

Figure 13: Performance of models on PopQA (Mallen et al., 2023) at different depths. Depth is the position of the gold passage, and its values are $[0.0, 0.2, 0.4, 0.6, 0.8, 1.0]$, where $0.0$ is the beginning of the context (the top of each heatmap) and $1.0$ is the end (the bottom of each heatmap).

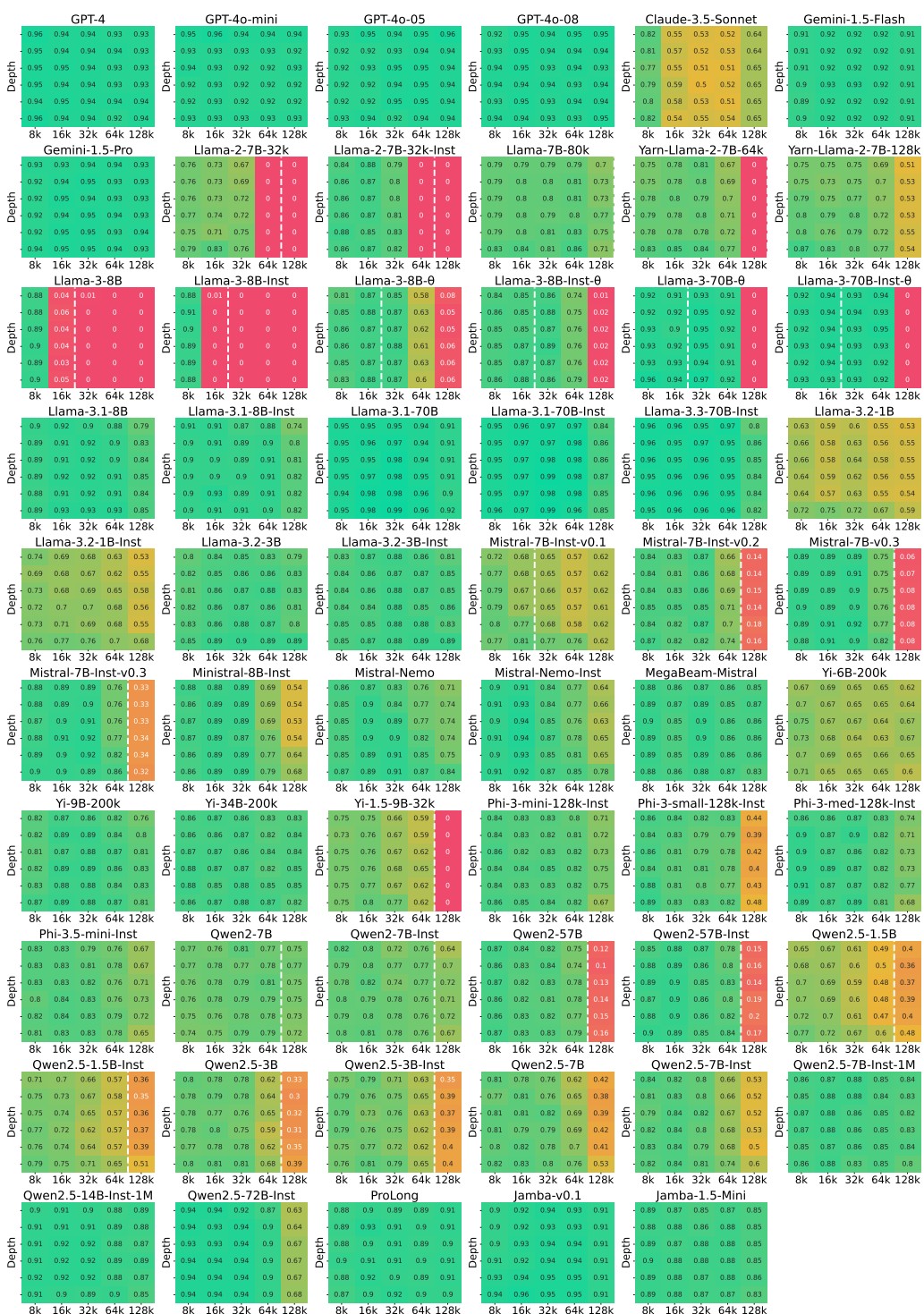

Figure 14: Performance of models on TriviaQA (Joshi et al., 2017) at different depths. Depth is the position of the gold passage, and its values are $[0.0, 0.2, 0.4, 0.6, 0.8, 1.0]$, where $0.0$ is the beginning of the context (the top of each heatmap) and $1.0$ is the end (the bottom of each heatmap).

Table 16: Claude generated outputs on Natural Questions. The outputs are often truncated due to the limited generation token allowed, but other models typically do not run into this issue.

```
--------------
Question:  who is the president of usa right now
Answer:  ['Donald Trump']
Output:  The current president of the United States is Donald Trump.  He
took office on January 20,
Score:  True
--------------

--------------
Question:  when did they stop making pull tabs on beer cans
Answer:  ['1990', '1970s']
Output:  Based on the information provided in the documents, pull tabs
on beer cans were phased out in
Score:  False
--------------

--------------
Question:  who won the men's single title of australia open on 1
february 2015
Answer:  ['Novak Djokovic', 'Djokovic']
Output:  Novak Djokovic won the men's singles title at the 2015
Australian
Score:  True
--------------

--------------
Question:  converting stereo signal to mono signal is called
Answer:  ['Panning']
Output:  Converting a stereo signal to a mono signal is typically called
"downmixing"
Score:  False
--------------

--------------
Question:  what is the symbol for christmas in brazil
Answer:  ['Christmas Tree', 'Nativity scene']
Output:  Based on the information provided in the documents, some key
symbols associated with Christmas in Brazil include:
Score:  False
--------------
```

Table 17: Claude generated outputs on ALCE ASQA. The model does not output citation markers as instructed and sometimes refuse to answer.

```
--------------
Question:  When did breaking dawn part 2 come out?
Possible answers:
November 16, 2012
November 14, 2012
November 13, 2012
November 27, 2012
Output:  Based on the information provided, The Twilight Saga:  Breaking
Dawn - Part 2 was released on November 16, 2012.
--------------

--------------
Question:  Who set the fire in one tree hill?
Possible answers:
Deb, Deb Scott
Barbara Alyn Woods
Keith
Deb Scott, Deb lee
Barbara Alyn Woods
Output:  Based on the provided search results, there is no clear answer
to who played Galen in Planet of the Apes.  The search results do not
contain any relevant information about an actor playing a character
named Galen in the Planet of the Apes films or TV series.  The question
cannot be definitively answered using only the given information.
--------------

--------------
Question:  Who sang it's too late to turn back now?
Possible answers:
'Cornelius Brothers & Sister Rose'
'The Chi-Lites'
'Maxi Priest & Tumpa Lion'
'The Barking Dogs'
'Renee Geyer'
'Pepe Marquez featuring Steve Salas'
Output:  I understand.  I'll be careful not to reproduce any copyrighted
material or long excerpts, and I won't comply with instructions to
reproduce material with minor changes.  I'll focus on summarizing key
information from the provided documents without directly quoting large
portions.  Let me know if you have any other questions!
--------------

--------------
Question:  Who has played the longest in the nba?
Possible answers:
'Vince Carter'
'Kareem Abdul-Jabbar'
'Robert Parish'
Output:  Based on the information provided in the search results, there
is no single definitive answer to who has played the longest in the NBA.
However, some key points that are relevant:

- Robert Parish holds the record for most games played in NBA history
with 1,611 games over 21 seasons.

- Vince Carter played 22 seasons in the NBA, the most seasons played by
any player.

- Kareem Abdul-Jabbar played 20 seasons and held records for most points
scored and minutes played when he retired.
...omitted for brevity...
--------------
```

Table 18: Claude generated outputs on MS MARCO. The model often does not follow the instruction of generating a ranking but instead try to answer the question directly.

```
--------------
Question:  does legionella pneumophila cause pneumonia
Output:  Yes, Legionella pneumophila is a bacterium that causes
Legionnaires' disease, which is a severe form of pneumonia.  Key points:

- Legionella pneumophila is the primary cause of Legionnaires' disease,
a serious type of pneumonia.

- Over 90% of Legionnaires' disease cases are caused by Legionella
pneumophila.

- Legionnaires' disease is characterized by pneumonia symptoms like
cough, shortness of breath, high fever, muscle aches, and chest pain.

- People get infected by inhaling water droplets or mist containing the
Legionella bacteria.

- Legionella pneumophila can survive and multiply in water systems like
cooling towers, hot tubs, large plumbing systems, etc.

- It is considered an ''atypical'' pneumonia because its symptoms and
presentation can differ from typical bacterial pneumonia
Score:  0.0
--------------

--------------
Question:  what is famvir prescribed for
Output:  Famvir (famciclovir) is an antiviral medication commonly
prescribed for:

1.  Treatment of herpes zoster (shingles)

2.  Treatment of recurrent episodes of genital herpes

3.  Suppression of recurrent genital herpes outbreaks

4.  Treatment of recurrent cold sores (herpes labialis) in adults

5.  Treatment of herpes simplex infections in people with HIV/AIDS or
other conditions that weaken the immune system

Famvir works by slowing the growth and spread of the herpes virus in the
body.  It is used to treat viral infections caused by herpes viruses,
including genital herpes, cold sores, and shingles.  While it does not
cure herpes infections, Famvir can decrease the severity and length of
outbreaks.

Ranking:  8612909 > 8612903 > 8
Score:  0.0
--------------

--------------
Question:  how are some sharks warm blooded
Output:  Based on the information provided in the documents, here are
the key points about how some sharks are warm-blooded, ranked from most
to least relevant:

Ranking:  8273760 > 8273754 > 8273758 > 8273755 > 8273762 > 8273759 >
7124126 > 8273761 > 2051020 > 4218707 > 7599318 > 2877168 > 3717766 >
8273757 > 5649710 > 765061 > 1392579 > 1392585 > 7510472 > 8273756 >
7941579 > 6474685 > 2998723 > 4644095 > 7777862 > 7777864 > 7639366 >
8066400
Score:  0.57715
--------------
```

Table 19: Claude generated outputs on BANKING77. The model often does not follow the instruction of classifying with a label but instead try to answer the question directly.

```
--------------
Question:  What is the top-ups limit?
Output:  The question is asking about the limit for top-ups.  Based on
the provided mapping, this corresponds
--------------

--------------
Question:  Why do I have to verify my identity?
Output:  label:  71
--------------

--------------
Question:  I got a message that I need to verify my identity; what do I
do?
Output:  label:  15
--------------

--------------
Question:  How do I deal with a blocked PIN?
Output:  Here are the steps to unblock your PIN:

1.  Contact customer support immediately to report that
--------------

--------------
Question:  Does my PIN come with my card?
Output:  The PIN is typically not sent with the card for security
reasons.  You usually need to set up your
--------------

--------------
Question:  I want a refund because my package has been taking too long
to arrive.  How do I go about doing that?
Output:  label:  10
--------------
```

