# OpenReview forum: "HELMET: How to Evaluate Long-context Models Effectively and Thoroughly"
_ICLR.cc/2025/Conference — ICLR 2025 Poster_

### Official Review · Reviewer_HTGH · 2024-11-01

**Soundness:** 3
**Presentation:** 3
**Contribution:** 3
**Rating:** 6
**Confidence:** 4

**Summary:**

The paper proposes HELMET, a benchmark designed to evaluate long-context language models across seven application-focused categories, addressing issues such as inadequate dataset length, noisy evaluation metrics, and inconsistencies in current benchmarks. Through empirical evaluation on 51 models, the authors argue that HELMET offers better differentiation among models compared to traditional synthetic tasks and demonstrates the inadequacy of simple benchmarks in predicting real-world performance.

**Strengths:**

- HELMET covers diverse tasks such as retrieval-augmented generation, passage re-ranking, and long-document QA, providing a comprehensive test bed for evaluating the full capabilities of long-context models.
- By introducing controllable length settings and using model-based metrics instead of n-gram matching, HELMET offers a better reflection of human judgments and real-world performance.
- The authors evaluate 51 models, providing valuable insights into how different architectures and model sizes handle long-context tasks.

**Weaknesses:**

- While HELMET’s application-oriented tasks are extensive, they may not fully capture long-context models’ capabilities in highly specific domains like legal or medical texts, limiting its applicability in niche areas.
- The heavy reliance on closed models such as GPT-4 for comparison leaves open questions about the efficacy of HELMET in an entirely open-source setting, which may limit reproducibility for some researchers.

**Questions:**

- How well does HELMET handle variations in domain-specific tasks, such as medical or financial documents?
- Could open-source models trained on synthetic datasets achieve comparable results with additional tuning on HELMET's diverse tasks?

---

> ### Author Response · Authors · 2024-11-18
> **Rebuttal**
>
> We thank the reviewer for the helpful feedback, and we are encouraged that the reviewer finds our benchmark to be diverse and reliable, and our insights valuable. The concerns are addressed below:
>
> > [W1, Q1] may not fully capture long-context models’ capabilities in highly specific domains like legal or medical texts, limiting its applicability in niche areas… How well does HELMET handle variations in domain-specific tasks, such as medical or financial documents?
>
> HELMET was designed to be a general-purpose benchmark, but it does include a highly specialized domain of law, as Multi-LexSum is a legal document summarization dataset.
> We show in Figure 9 that general-purpose datasets can also have a high correlation with performance on more specific domains. For instance, Multi-LexSum has a Spearman $\rho = 0.91$ with the $\infty$Bench Sum  dataset (books).
> While it would be interesting to consider other domains as well, a  general-purpose benchmark provides a good signal of how the models may perform in more niche areas.
> We leave such explorations for future work.
>
> > [W2] heavy reliance on closed models such as GPT-4 for comparison
>
> Only 3 out of the 21 datasets from HELMET rely on GPT-4 for evaluation (Multi-LexSum, InfiniteBench Sum, and NarrativeQA); the others either rely on automatic metrics or open-sourced models. Thus, most of HELMET are evaluated in entirely open-source settings.
> Furthermore, we chose to use closed-source models because (1) they are easier to run than large open-source models (e.g., Llama 70B) in practice since they are only API calls and do not require multiple GPUs; (2) using closed-source models for evaluation is a widely-accepted standard practice in the community, such as AlpacaEval2 [1] and WildBench [2]; (3) closed-source models are much better judges than open-source models overall [3][4].
>
> > [Q2] Could open-source models trained on synthetic datasets achieve comparable results with additional tuning on HELMET's diverse tasks?
>
> The use of synthetic data in long-context training is an interesting research topic and we believe it could hold huge potential in terms of boosting model performance,
> However, we do not recommend directly fine-tuning the models on HELMET tasks’ training sets, as this likely will lead to overfitting on specific tasks.
>
> [1] Dubois et al., 2024. Length-Controlled AlpacaEval: A Simple Way to Debias Automatic Evaluators
>
> [2] Lin et al., 2024. WildBench: Benchmarking LLMs with Challenging Tasks from Real Users in the Wild
>
> [3] Zheng et al., 2023. Judging LLM-as-a-Judge with MT-Bench and Chatbot Arena
>
> [4] Zeng et al., 2024. Evaluating Large Language Models at Evaluating Instruction Following

---

> ### Author Response · Authors · 2024-11-22
> **Reminder for Paper Discussion**
>
> Dear reviewer,
>
> As we approach the end of the discussion period, we would greatly appreciate your input on the paper. We hope our responses and additional results address your concerns and welcome any further questions or suggestions.

---

> ### Author Response · Authors · 2024-12-02
> **Reminder for Paper Discussion**
>
> Dear Reviewer,
>
> As the deadline for the discussion period approaches, we hope that we have addressed your concerns of HELMET's application to domain-specific tasks and the use of closed-source models during evaluation. We would greatly appreciate it if you could respond to our rebuttal. Please let us know if you have any other questions or concerns!

---

### Official Review · Reviewer_TJGb · 2024-11-03

**Soundness:** 3
**Presentation:** 3
**Contribution:** 3
**Rating:** 6
**Confidence:** 4

**Summary:**

The paper presents HELMET, a benchmark for evaluating long-context language models (LCLMs) that try to address limitations in existing evaluations, which often rely on synthetic tasks lacking real-world applicability. HELMET includes 7 diverse, application-centric tasks and supports input lengths up to 128k tokens. Through evaluating 51 LCLMs, the authors demonstrate that synthetic tasks are poor predictors of downstream performance, different task categories exhibit distinct trends, and open-source models significantly lag behind closed-source models on complex tasks requiring reasoning over long contexts. They advocate for holistic evaluation across diverse tasks to gain a comprehensive understanding of LCLM capabilities.

**Strengths:**

* The paper attempts to provide a standardized, holistic benchmark for LCLMs, whose adoption can potentially improve consistency and reliability in model evaluation and comparison.
* The evaluation is extensive -- 51 LCLMs across multiple dimensions, tasks, input lengths, and model types (open-, closed-source)
* The paper provide some valuable findings and insights into the performance of LCLMs, e.g. the limitations of synthetic tasks as predictors of real-world performance and where the performance gaps are between open- and closed-source models. This can guide future research and model development.

**Weaknesses:**

* The authors observe that on existing benchmarks like RULER and ∞BENCH, smaller models (e.g., Llama-8B) sometimes outperform larger ones (e.g., Gemini Pro, Llama-70B), and they conclude that these benchmarks are unreliable because they do not reflect human expectations that larger models should perform better. This reasoning may be premature and somewhat biased. It's possible that the larger models genuinely underperform on these benchmarks due to specific issues, such as overfitting, architectural limitations, or difficulties in handling certain tasks. The benchmarks might be accurately capturing these performance discrepancies. Dismissing unexpected results as benchmark unreliability without thoroughly investigating the underlying causes undermines the validity of the authors' argument. More analysis considering both the possibility of model issues and benchmark limitations would strengthen the conclusions.
* While the paper introduces model-based evaluation metrics using 4o to address the unreliability of traditional metrics like ROUGE, it provides limited details on how these metrics were validated against human judgments. Including more detailed results or analysis of human-model agreement would strengthen the validity of the evaluation methodology.
* Although the paper critiques existing benchmarks, it could offer more in-depth analysis demonstrating how HELMET improves over them in practice. Figure 1 seems to be the only place where a direct comparison is shown. Conducting more direct comparisons of model rankings or performance differences on HELMET and existing benchmarks and providing concrete evidence of HELMET's advantages would strengthen the paper's arguments.

**Questions:**

1. In your analysis, you conclude that existing benchmarks like RULER and ∞BENCH are unreliable because larger models sometimes perform worse than smaller ones, which contradicts human expectations. Could you elaborate on why you attribute these unexpected results to benchmark unreliability rather than potential issues with the larger models themselves? Did you investigate alternative explanations for the performance discrepancies?
2. Do you have any results from human evaluation that validates the model-based evaluation metrics? What were the human-model agreement rates? Were there any notable discrepancies between the human judgments and model-based evaluations?
3. Other than RAG, which types of tasks in HELMET are compatible with the base model without instruction following capabilities?

---

> ### Author Response · Authors · 2024-11-18
> **Rebuttal 1/2**
>
> We thank the reviewer for the helpful feedback, and we are encouraged that the reviewer finds our evaluation extensive and our findings valuable. We address the concerns below:
>
> > [W1, Q1] More analysis considering both the possibility of model issues and benchmark limitations… why attribute these unexpected results to benchmark unreliability rather than potential issues with the larger models themselves? Did you investigate alternative explanations for the performance discrepancies?  [W3] more in-depth analysis demonstrating how HELMET improves over them in practice… more direct comparisons of model rankings or performance differences on HELMET and existing benchmarks
>
> Please see our general response.
>
> > [W2, Q2] Do you have any results from human evaluation that validates the model-based evaluation metrics? What were the human-model agreement rates? Were there any notable discrepancies between the human judgments and model-based evaluations?
>
> We conducted human validation of our model-based evaluation in Appendix B.5. In the rebuttal, we added a more rigorous and comprehensive human evaluation of our methods, as detailed below.
>
> We first check the key point generation procedure, where we ask the LLM to generate key points from a long summary. We manually check 105 key points originating from 25 Multi-LexSum human-written summaries and find all claims to be factually correct and a major claim in the summary. We did notice one instance with the model excluding a possible key point, but overall, we find GPT-4o to be reliable for key point generation, which also aligns with previous findings [1][2].
>
> Then, we verify if the judge can correctly identify if a key point is supported by the generated summary for the recall score. We randomly sample 10 generated summaries from Multi-LexSum and $\infty$Bench Sum each (generated by Llama-3.1-70B-Instruct and Gemini-1.5-Pro), and manually check the five key point evaluations for each summary (totaling 100 checks). We find a Cohen $\kappa = 0.76$ for $\infty$Bench Sum and $\kappa = 0.72$ for Multi-LexSum, suggesting substantial agreement.
> Qualitatively, we find that most of the disagreements between humans and the model arise from the partially supported cases. For instance, the key point may include specific details, such as the names of government departments or Court Justices’ names, that are not explicitly mentioned in the generated summary, and the model judge is typically more lenient about the exclusion of these small details while humans are more strict. However, this is also subjective to the preference of the human.
>
> We perform a similar analysis to check if the judge can judge the precision score correctly, and we find $\kappa = 0.91$ for $\infty$Bench Sum and $\kappa = 0.83$ for Multi-LexSum, suggesting near-perfect agreement. We notice a similar trend as before – most judgments are straightforward and agreed upon, but there are some nuanced cases where humans may pay more attention to specific details whereas the model judge is more lenient.
>
> Finally, we manually evaluate the fluency of all the generated summaries from the previous analysis, and we find that we always agree with the model judgment. This is likely due to the clear definitions of fluency in our prompt (shown in Table 10) and the easy nature of the judgment.  Thus, we believe that our model-based evaluations are reliable and correlate well with human judgments.
> We have updated the PDF with human evaluation in Section 2.2 and Appendix B (highlighted in red).
>
> [1] Kamoi et al., 2023. WiCE: Real-World Entailment for Claims in Wikipedia.
>
> [2] Gao et al., 2023. Enabling Large Language Models to Generate Text with Citations

---

> > ### Comment · Reviewer_TJGb · 2024-11-27
> >
> > I would like to thank the authors for the details response and for addressing the concerns raised. The general response does also help clarify the reasoning behind attributing the unexpected ranking to existing benchmarks. I will maintain my current scores.

---

> ### Author Response · Authors · 2024-11-18
> **Rebuttal 2/2**
>
> > [Q3] which types of tasks in HELMET are compatible with the base model without instruction following capabilities?
>
> Base models are compatible with all tasks in HELMET due to the few-shot demonstrations. We show this in Table 8—base models benefit significantly from just two in-context learning examples, and achieve non-trivial performance comparable to other instruction-tuned models. We believe that this practice is a better reflection of the base model’s long-context abilities and will be useful in model development.

---

> ### Author Response · Authors · 2024-11-22
> **Reminder for Paper Discussion**
>
> Dear reviewer,
>
> As we approach the end of the discussion period, we would greatly appreciate your input on the paper. We hope our responses and additional results address your concerns and welcome any further questions or suggestions.

---

### Official Review · Reviewer_1MDc · 2024-11-03

**Soundness:** 3
**Presentation:** 3
**Contribution:** 3
**Rating:** 6
**Confidence:** 3

**Summary:**

The paper introduces a new benchmark called HELMET, which is designed to comprehensively evaluate the performance of long-context language models (LCLMs). Current LCLM evaluations largely rely on synthetic tasks, like Needle-in-a-Haystack (NIAH), or arbitrary subsets of some datasets. However, these methods present issues such as high noise, insufficient coverage of downstream applications, inadequate dataset lengths, and unreliable metrics. HELMET aims to address these shortcomings by expanding task diversity across seven application-centric categories (including long-document QA, citation-based generation, etc.), supporting controllable input lengths up to 128k tokens, and implementing model-based evaluations for more reliable results. Through testing 51 LCLMs, this study finds that synthetic tasks are poor predictors of downstream performance, open-source models fall behind closed-source models on complex long-context tasks, and there is low correlation among task categories, highlighting the need for multi-dimensional LCLM evaluation .

**Strengths:**

1.	**Diverse Task Design**: HELMET includes seven categories of tasks, enhancing the representativeness of LCLMs in real applications.

2.	**Support for Ultra-Long Inputs**: This benchmark accommodates input lengths over 128k tokens, making it suitable for evaluating the long-context capabilities of frontier models.

3.	**Reliable Model-Based Evaluation**: HELMET’s evaluation metrics reflect human judgment better than traditional n-gram matching, offering more reliable model ranking.

4.	**Compatibility with Base Models**: The benchmark allows evaluations of base models that haven’t undergone instruction fine-tuning, broadening LCLM applicability.

**Weaknesses:**

1.	**High Complexity**: With multiple tasks and model comparisons involved, HELMET’s setup and evaluation process is intricate and demands considerable effort from researchers.

2.	**Low Correlation Among Some Tasks**: The low correlation between different tasks may make it challenging to assess a model’s overall long-context handling ability if it performs exceptionally in only certain tasks.

1. **High Resource Consumption**: Running the full suite of HELMET tasks is time-intensive. It would be beneficial to identify a few key subtasks that can maintain consistency with the results of full testing, allowing for time-saving evaluations.

**Questions:**

Please address the weaknesses in the previous section.

---

> ### Author Response · Authors · 2024-11-18
> **Rebuttal**
>
> We thank the reviewer for the detailed comments, and we are encouraged that the reviewer finds HELMET to be diverse and reliable. The concerns are addressed below:
>
> > [W1] High Complexity… ​​demands considerable effort from researchers
>
> We kindly disagree with the reviewer. (1) The complex nature of real-world long-context applications requires a holistic and diverse evaluation, which we spent a significant effort on. We show in Figure 1 that focusing on narrow domains in evaluation leads to inconsistent and unintuitive comparisons. (2) The complexity of designing the benchmark does not hinder its usability. On the contrary, we spent meticulous effort in building HELMET so that developers could evaluate diverse tasks by using just one benchmark. We also built an easy-to-use codebase where all experiments can be reproduced by one command, which we will release with the paper.  (3) We also recommended using the RAG subset for fast development, which we supported with rigorous correlation analysis across different applications.
>
> > [W2] Low Correlation Among Some Tasks… challenging to assess a model’s overall long-context handling ability
>
> A major argument of our work is the necessity of holistic evaluation of long-context language models across diverse domains—long-context abilities cannot be compressed into one number or tested by a single task, such as NIAH or perplexity (Section 3.2). Thus, HELMET offers a comprehensive evaluation across different axes that will paint a better picture of the model’s long-context performance compared to previous benchmarks.
>
> > [W3] High Resource Consumption
>
> Thank you for pointing it out! We recommended using the synthetic recall and RAG tasks during model development for faster iterations (Section 3.1). But we also argue that compute resources required by HELMET are negligible compared to long-context training (e.g., it only takes 1 GPU for running evaluation for an 8B model but takes more than 8 GPUs for days to fine-tune it on 128K).
> Additionally, we show the correlation across all tasks in Figure 9—we observe that certain datasets may be used during development as a proxy for the entire category. For example, BANKING77 achieves a high correlation with the rest of the ICL tasks while InfBench MC also achieves a high correlation with the Long-Document QA performances. We will include these recommendations in the revision.
> Furthermore, we optimize our public code repo with tools such as FlashAttention and vLLM to reduce the cost of running HELMET. Consequently, the entire HELMET evaluation of an 8B model at 128K input lengths finishes within 16 hours on one H100 GPU (most datasets take less than 30 minutes to run except for the long generation tasks). The Recall set can be completed in less than 90 minutes for faster development.

---

> ### Author Response · Authors · 2024-11-22
> **Reminder for Paper Discussion**
>
> Dear reviewer,
>
> As we approach the end of the discussion period, we would greatly appreciate your input on the paper. We hope our responses and additional results address your concerns and welcome any further questions or suggestions.

---

> > ### Comment · Reviewer_RMnk · 2024-11-26
> >
> > Thank you for the response. It answers my questions. Good work. **I'd like to raise the score to 7**. But due to the coarse granularity of rating. I will keep my score as 6.

---

> > ### Comment · Reviewer_1MDc · 2024-11-28
> >
> > Thanks for the authors' responses to the complexity of the benchmark as well as the resource consumption. Considering the overall techincal contributions of the paper, I would keep my evaluation score at 6.

---

### Official Review · Reviewer_RMnk · 2024-11-04

**Soundness:** 3
**Presentation:** 4
**Contribution:** 4
**Rating:** 6
**Confidence:** 5

**Summary:**

This paper constructs a comprehensive benchmark to test LLMs' long context abilities. It covers various types of tasks such as RAG, ICL, LongQA, Retrieval, Re-rank and so on.  The used prompts and evaluation metrics and carefully designed to ensure both IFT models and base models can give predictions. This benchmark also evaluates most commonly recognized LLMs and accordingly provides insights about LLMs' long context performance.

**Strengths:**

1: The benchmark is comprehensive. It covers most real-world long context use cases.

2: The investigation of performance correlation among all task types are insightful. It provides a new perspective to understand LLMs' long context ability.

3: The improvement to prompting strategy and evaluation method effectively stabilizes the evaluation results.

**Weaknesses:**

1: The so called "expected" ranking of LLMs is a bit subjective.

2: Lack of some deep analysis to interesting results, such as why the json-kv task has higher correlation with re-rank than RAG or LongQA

3: The RoPE scaling settings are not suitable for 128k/64k testing. With ABF, usually, the scaling factor should be at least 2x the target extension ratio. With 8k context, Llama3 should use at least a scaling factor of 32 for 128k testing.

**Questions:**

1: Figure 2 is missing?

2: What is the value for 'depth' in Figure 11? From top to the bottom, is the key information located at the beginning of the context to the tail of the context?

3: Gemma series have a unique attention head dimension of 256 rather than 128. It might have interesting impact on the long context things. It would be better to have results with Gemma series as the tested models.

---

> ### Author Response · Authors · 2024-11-18
> **Rebuttal**
>
> We thank the reviewer for the detailed and helpful comments. We are encouraged that the reviewer finds HELMET comprehensive and that our findings are insightful. The concerns are addressed below:
>
> > [W1] The so called "expected" ranking of LLMs is a bit subjective.
>
> Please refer to our general response.
>
> > [W2] Lack of some deep analysis to interesting results… why the json-kv task has higher correlation with re-rank than RAG or LongQA
>
> To clarify, we find, in Figure 8, that RAG overall achieves a higher correlation with Re-rank ($\rho = 0.9$) than JSON KV ($\rho=0.83$). LongQA also observes a high correlation with Re-rank ($\rho = 0.89$; Figure 9). This is intuitive because RAG requires the model to leverage retrieved passages, similar to passage re-ranking.
>
> We agree that there are more interesting possible analyses in this setting, and we dive into them in Appendix E, where we analyze the correlation between individual datasets, challenges in positional embeddings, the lost-in-the-middle problem, and comparison between base and instruction-tuned models.
> Furthermore, we have since added more qualitative analysis into the failure of specific models; for example, we find some closed-source models, such as Claude, tend to not follow the instructions in the ICL tasks. The updated PDF includes these new analyses in Appendix E.6.
> We hope the release of HELMET will enable more interesting analyses and model development in the field of long contexts. We plan to release an easy-to-use code repository for the community to use.
>
> > [W3] With 8k context, Llama3 should use at least a scaling factor of 32 for 128k testing.
>
> Thank you for the suggestion, we show the results on HELMET at 128K input length for the Llama 3 models with RoPE Theta base set to 8M (scaling factor of 16) and 16M (scaling factor of 32) below. The new results are highlighted in **bold**:
>
> |                      | Recall | RAG | ICL | Cite | Re-rank | LongQA | Summ | Ours |
> |----------------------|--------|-----|-----|------|---------|--------|------|------|
> | Llama-3-8B-Inst-8Mθ    | 0.0    | 0.2 | 0.2 | 0.1  | 0.0     | 5.7    | 11.2 | 2.5  |
> | **Llama-3-8B-Inst-16Mθ**  | 0.0    | 1.5 | 3.4 | 0.9  | 0.0     | 9.2    | 13.2 | 4.0  |
> | Llama-3-70B-Inst-8Mθ   | 0.0    | 0.0 | 0.0 | 0.0  | 0.0     | 9.2    | 12.0 | 3.0  |
> | **Llama-3-70B-Inst-16Mθ** | 0.0    | 0.0 | 0.0 | 0.0  | 0.0     | 7.3    | 14.9 | 3.2  |
> | Llama-3.1-8B-Inst    |   99.4 | 69.1 | 69.8 | 35.4 |    58.7 |   24.6 | 23.2 | 54.3 |
> | Llama-3.1-70B-Inst   |   99.9 | 73.0 | 71.6 | 44.5 |    73.3 |   31.5 | 27.7 | 60.2 |
>
> We find that the model performances do improve, but the absolute changes are still relatively small, and the models degenerate on most tasks, remaining at 0% performance. We will include these results in the final revision.
>
> > [Q1] Figure 2 is missing?
>
> Figure 2 is located at the top of Page 6.
>
> > [Q2] What is the value for 'depth' in Figure 11?
>
> Depth is the location of the gold passage (or the location of the needle) that contains the answer to the question. From top to bottom, the gold passage moves from the very start of the context (depth=0.0) to the very end of the context (depth=1.0). The depth is evenly spaced out and takes on the values of {0.0, 0.2, 0.4, 0.6, 0.8, 1.0}.
>
> > [Q3] It would be better to have results with Gemma series as the tested models.
>
> Thank you for the suggestion, we show Gemma-2 results on HELMET at 128K input length below, where the new results are highlighted in **bold**:
>
> |                    | Recall | RAG  | ICL  | Cite | Re-rank | LongQA | Summ | Ours |
> |--------------------|--------|------|------|------|---------|--------|------|------|
> | **Gemma-2-9B**         | 0.00   | 0.00 | 0.00 | 0.00 | 0.00    | 1.36   | 0.00 | 0.19 |
> | **Gemma-2-9B-Inst**    | 0.00   | 0.00 | 0.00 | 0.00 | 0.00    | 13.18  | 0.00 | 1.88 |
> | **Gemma-2-9B-Inst-θ**  | 0.00   | 2.29 | 0.20 | 0.00 | 0.00    | 8.63   | 0.00 | 1.59 |
> | **Gemma-2-27B**        | 0.00   | 0.42 | 0.40 | 0.39 | 0.00    | 2.03   | 0.00 | 0.46 |
> | **Gemma-2-27B-Inst**   | 0.00   | 0.25 | 0.00 | 0.23 | 0.00    | 6.36   | 0.32 | 1.02 |
> | **Gemma-2-27B-Inst-θ** | 0.00   | 1.13 | 5.40 | 0.70 | 0.00    | 1.18   | 0.00 | 1.20 |
> | Llama-3.1-8B-Inst    |   99.4 | 69.1 | 69.8 | 35.4 |    58.7 |   24.6 | 23.2 | 54.3 |
> | Llama-3.1-70B-Inst   |   99.9 | 73.0 | 71.6 | 44.5 |    73.3 |   31.5 | 27.7 | 60.2 |
>
> We test both the base models and the instruction-tuned models (θ denotes changing the RoPE Theta base from 10k to 320k). Since the Gemma-2 models were trained with a context window of 8K tokens, they often degenerate on long-context tasks. We will include these results and analyses in the final revision.

---

> ### Author Response · Authors · 2024-11-22
> **Reminder for Paper Discussion**
>
> Dear reviewer,
>
> As we approach the end of the discussion period, we would greatly appreciate your input on the paper. We hope our responses and additional results address your concerns and welcome any further questions or suggestions.

---

### Author Response · Authors · 2024-11-18
**General Response Rebuttal 1/2**

We thank all reviewers for their helpful feedback. We are encouraged that all reviewers found HELMET to be a comprehensive and practical benchmark and our analyses to be insightful. A common concern is addressed below:

> the “expected” ranking of models can be subjective and why we attribute the unexpected results to benchmark unreliability rather than issues with the models. what are alternative explanations for the performance discrepancies? more direct comparisons of model rankings or performance differences on HELMET and existing benchmarks

We attribute the discrepancy in model rankings across benchmarks to the following reasons:

1. **Previous work finds that larger models perform better than smaller models at long context.** The Gemini 1.5 report observes a trend of larger models performing better at long contexts—the larger models beating the smaller models both qualitatively and quantitatively across the Gemini, GPT, and Claude model families (Section 5.2, Tables 4 and 5) [4]. Although the evaluation suite is not publicly available, the results align with our intuition, which is commonly believed in other use cases of language models. Since we do not have access to a ground truth ranking of models and acquiring one via human evaluation is nearly impossible, relying on such a commonly-accepted intuition is reasonable.

2. **$\infty$Bench still relies on n-gram matching metrics for QA and summarization evaluation, which have been shown to be unreliable and noisy** [1][2][3].
We discuss this in-depth in Section 2.2. In Figure 2, we observe that ROUGE-L scores cannot effectively distinguish different models at different input lengths, but our model-based evaluation identifies improvements and degeneration with increasing input lengths and distinguishes models of different capabilities.
From additional qualitative analysis, we found that model-based evaluations are better at penalizing decoding degenerations as well. For instance, on $\infty$Bench Sum, a model generated a summary with the sentence “The author’s object is to show that the study of grammar is necessary part of education” repeated hundreds of times. This summary receives a ROUGE F1 score of 12.3 while our model-based metric identifies it as incoherent and assigns it a score of 0. We will offer more qualitative analysis in the revision.

3. **HELMET improves prompting strategies, such as using in-context learning examples.**
In Table 7, we show that using better instruction and in-context learning results in better performance for many models, which is a more accurate representation of how these models would be used in practice. This is especially important for smaller models, which degenerate without the in-context learning examples. Ablations of the improvement from ICL examples are shown in Table 8.

4. **Previous benchmarks lack diverse and challenging applications.** In benchmarks like RULER, many models already achieve near-perfect scores on certain synthetic tasks, which cannot provide meaningful signals and exhibit low correlations with real-world applications (Section 3.1, Figures 3 and 4). On the other hand, HELMET better distinguishes frontier models by introducing diverse and complex tasks such as generation with citations (Section 3.3).

---

> ### Author Response · Authors · 2024-11-19
> **General Response Rebuttal 2/2**
>
> 5. **In practice, these improvements result in HELMET more accurately reflecting model performance.**
> We take a closer look at a subset of open-sourced models’ ranking and performance on HELMET and $\infty$Bench below. The new results added during the rebuttal are highlighted in **bold**.
>
> | Model              | HELMET |   | Model              | $\infty$Bench |
> |--------------------|--------|---|--------------------|----------|
> | Llama-3.1-70B-Inst |   49.3 |   | Llama-3.1-8B-Inst  |     46.7 |
> | Llama-3.1-8B-Inst  |   47.0 |   | Llama-3.1-70B-Inst |     43.7 |
> | Llama-3.1-70B      |   41.3 |   | **Yi-34B-200k**        |     43.1 |
> | Yi-34B-200k        |   38.3 |   | **Llama-3.1-70B**      |     38.6 |
> | Llama-3.2-3B-Inst  |   36.9 |   | **Yi-9B-200k**         |     37.6 |
> | Llama-3.1-8B       |   35.6 |   | **Llama-3.1-8B**       |     35.7 |
> | Yi-9B-200k         |   33.0 |   | **Yi-6B-200k**         |     32.0 |
> | Llama-3.2-3B       |   31.9 |   | **Llama-3.2-3B-Inst**  |      2.8 |
> | Yi-6B-200k         |   26.3 |   | **Llama-3.2-1B-Inst**  |      2.6 |
> | Llama-3.2-1B-Inst  |   24.6 |   | **Llama-3.2-1B**       |      2.0 |
> | Llama-3.2-1B       |   21.2 |   | **Llama-3.2-3B**       |      1.8 |
>
> Other than the performance discrepancy between Llama-3.1-8B-Inst and Llama-3.1-70B-Inst, which we noted in Figure 1, we also see that $\infty$Bench shows that all Llama-3.2 models degenerate at long contexts (128K tokens).
> However, on HELMET, we find that the Llama-3.2 models, especially 3B-Inst, rank well against other open-source models. Upon qualitative examination, we find that the Llama 3.2 models are able to produce coherent and useful generations at long contexts with better prompting strategies from HELMET, such as adding in-context learning examples. Thus, HELMET provides a better reflection of how these models would be used in practice over previous benchmarks.
>
> We appreciate the reviewers for the helpful feedback, and we have updated the PDF with the additional discussions in the introduction and Appendix A (new edits are highlighted in red).
>
> [1] Deutsch et al., 2022. Re-examining system-level correlations of automatic summarization evaluation metrics
>
> [2] Goyal et al., 2023 News Summarization and Evaluation in the Era of GPT-3
>
> [3] Chang et al., 2024. Booookscore: A systematic exploration of book-length summarization in the era of llms
>
> [4] Gemini Team, Google, et al., 2024. Gemini 1.5: Unlocking multimodal understanding across millions of tokens of context.

---

### Meta-Review · Area_Chair_GCTK · 2024-12-23

**Metareview:**

This paper presents HELMET, a comprehensive benchmark for evaluating long-context language models (LCLMs) across seven application categories, with controllable lengths up to 128k tokens. The key claims include that synthetic tasks like needle-in-a-haystack (NIAH) are poor predictors of downstream performance, different task categories exhibit distinct uncorrelated trends, and open-source models lag significantly behind closed models on complex reasoning tasks. The paper's strengths include: providing a holistic evaluation framework encompassing diverse real-world tasks, demonstrating issues with current benchmarks through rigorous analysis, and offering practical insights through extensive evaluation of 51 models. While initial concerns were raised about the "expected" ranking of models being subjective and the heavy reliance on closed models for evaluation, the authors provided strong evidence and justification during rebuttal. They conducted additional experiments with different RoPE settings on Llama models, added comprehensive human validation of their model-based metrics showing high agreement (kappa > 0.8), and demonstrated that only 3 of 21 datasets rely on closed models for evaluation. For these reasons I vote to accept the paper!

**Additional Comments On Reviewer Discussion:**

The reviewers raised several key points that led to constructive discussion. Reviewer RMnk questioned the subjectivity of model rankings and RoPE configurations - the authors responded with additional experiments using different RoPE settings and Gemma models, leading RMnk to express interest in raising their score. Reviewer 1MDc raised concerns about complexity and resource requirements - the authors clarified that RAG tasks can be used for fast development and provided detailed runtime estimates. Reviewer TJGb questioned the attribution of unexpected rankings to benchmark unreliability - the authors justified this through detailed analysis of model-based metrics and human validation studies. Reviewer HTGH asked about domain-specific applications - the authors demonstrated high correlation between general and specialized domains like legal text. Overall, the discussion was highly productive, with authors providing comprehensive responses including new experimental results, human evaluations, and detailed compute analysis. Most reviewers explicitly acknowledged satisfaction with the responses, though maintained their original positive scores due to the coarse rating scale

---

### Decision · Program_Chairs · 2025-01-22

Accept (Poster)